# Learning from Reward-Free Offline Data:
# A Case for Planning with Latent Dynamics Models

**Vlad Sobal**[*1]  **Wancong Zhang**[*1]  **Kynghyun Cho**[1,2]  **Randall Balestriero**[3]
**Tim G. J. Rudner**[4]  **Yann LeCun**[1,5]

[1]New York University  [2]Genentech  [3]Brown University  [4]University of Toronto  [5]Meta – FAIR

🌐 Website  Code

## Abstract

A long-standing goal in AI is to develop agents capable of solving diverse tasks across a range of environments, including those never seen during training. Two dominant paradigms address this challenge: (i) reinforcement learning (RL), which learns policies via trial and error, and (ii) optimal control, which plans actions using a known or learned dynamics model. However, their comparative strengths in the offline setting—where agents must learn from reward-free trajectories—remain underexplored. In this work, we systematically evaluate RL and control-based methods on a suite of navigation tasks, using offline datasets of varying quality. On the RL side, we consider goal-conditioned and zero-shot methods. On the control side, we train a latent dynamics model using the Joint Embedding Predictive Architecture (JEPA) and employ it for planning. We investigate how factors such as data diversity, trajectory quality, and environment variability influence the performance of these approaches. Our results show that model-free RL benefits most from large amounts of high-quality data, whereas model-based planning generalizes better to unseen layouts and is more data-efficient, while achieving trajectory stitching performance comparable to leading model-free methods. Notably, planning with a latent dynamics model proves to be a strong approach for handling suboptimal offline data and adapting to diverse environments.

## 1 Introduction

How can we build a system that performs well on unseen combinations of tasks and environments? One promising approach is to avoid relying on online interactions or expert demonstrations, and instead leverage large collections of existing suboptimal trajectories without reward annotations [16, 32, 53]. Broadly, two dominant fields offer promising solutions for learning from such data: reinforcement learning and optimal control.

While online reinforcement learning has enabled agents to master complex tasks—from Atari games [44], Go [60], to controlling real robots [49]—it demands massive quantities of environment interactions. For instance, OpenAI et al. [49] used the equivalent of 100 years of real-time hand manipulation experience to train a robot to reliably handle a Rubik's cube. To address this inefficiency, offline RL methods [18, 33, 41] have been developed to learn behaviors from state–action trajectories with corresponding reward annotations. However, these methods typically train agents for a single task, limiting their reuse in other downstream tasks. To overcome this, recent work has explored learning behaviors from offline reward-free trajectories [32, 52, 53, 67]. This reward-free paradigm is particularly appealing as it allows agents to learn from suboptimal data and use the learned policy to solve a variety of downstream tasks. For example, a system trained on low-quality robotic interactions with cloth can later generalize to tasks like folding laundry [9].

---

* Equal contribution. Author ordering determined by coin flip.

39th Conference on Neural Information Processing Systems (NeurIPS 2025).

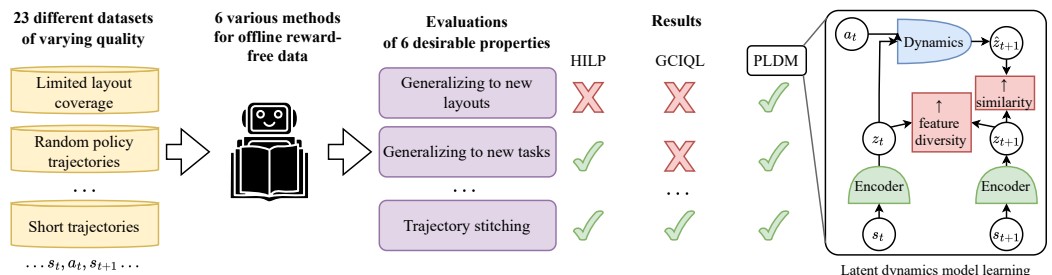

Figure 1: **Overview of our analysis.** We test six methods for learning from offline reward-free trajectories on 23 different datasets across several navigation environments. We evaluate for six generalization properties required to scale to large offline datasets of suboptimal trajectories. We find that planning with a latent dynamics model (PLDM) demonstrates the highest level of generalization. For a full comparison, see Table 1. **Right:** diagram of PLDM. Circles represent variables, rectangles – loss components, half-ovals – trained models.

Optimal control tackles challenge differently: instead of learning a policy function via trial and error, it plans actions using a known dynamics model [7, 63, 64] to plan out actions. Since real-world dynamics are often hard to specify exactly, many approaches instead learn the model from data [20, 71, 77]. This model-based approach has shown generalization in manipulation tasks involving unseen objects [17]. Importantly, dynamics models can be trained directly from reward-free offline trajectories, making this a compelling route [16, 56].

Despite significant advances in RL and optimal control, the role of pre-training data quality on reward-free offline learning remains largely unexplored. Prior work has primarily focused on RL methods trained on data from expert or exploratory policies [22, 76], without isolating the specific aspects of data quality that influence performance. In this work, we address this gap by systematically evaluating the strengths and limitations of various approaches for learning from reward-free trajectories. We assess how different learning paradigms perform under offline datasets that vary in both quality and quantity. To ground our study, we focus on navigation tasks — an essential aspect of many real-world robotic systems — where spatial reasoning, generalization, and trajectory stitching play a critical role. While this choice excludes domains such as manipulation, it offers a controlled yet challenging testbed for our comparative analysis.

Our contributions can be summarized as follows:

1. We propose two new navigation environments with granular control over the data generation process, and generate a total of *23 datasets* of varying quality;
2. We evaluate methods for learning from offline, reward-free trajectories, drawing from both reinforcement learning and optimal control paradigms. Our analysis systematically assesses their ability to learn from random policy trajectories, stitch together short sequences, train effectively on limited data, and generalize to unseen environment layouts and tasks beyond goal-reaching;
3. We demonstrate that learning a latent dynamics model and using it for planning is robust to suboptimal data quality and achieves the highest level of generalization to environment variations;
4. We present a list of guidelines to help practitioners choose between methods depending on available data and generalization requirements.

To facilitate further research into methods for learning from offline trajectories without rewards, we release code, data, environment visualizations, and more at `latent-planning.github.io`.

## 2 Related Work

**Reward-free offline RL** refers to learning from offline data that does not contain rewards in a task-agnostic way. The goal is to extract general behaviors from offline data to solve a variety of downstream tasks. One approach uses goal-conditioned RL, with goals sampled in similar manner as in Hindsight Experience Replay [1]. Park et al. [52] show that this can be applied to learn a goal-conditioned policy using IQL, as well as to learn a hierarchical value function. Hatch et al. [29] proposes using a small set of observations corresponding to the solved task to define the task and learn from reward-free data. Hu et al. [30], Yu et al. [79] propose to use labeled data to train a reward function, than label the reward-free trajectories. Zero-shot methods go beyond goal-reaching from

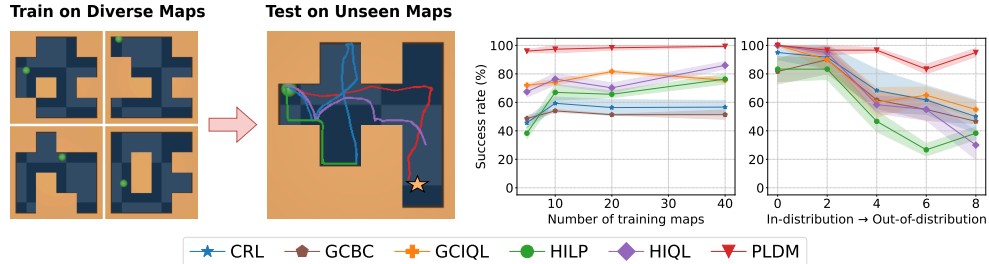

Figure 2: **Left:** We train offline goal-conditioned agents on trajectories collected in a subset of maze layouts (left), and evaluate on held out layouts, observing trajectories shown on the right. Only PLDM solves the task (see Figure 8 for more). **Right:** Success rates of tested methods on held-out layouts, as a function of the number of training layouts. Rightmost plot shows success rates of models trained on data from five layouts, evaluated on held-out layouts ranging from those similar to training layouts to out-of-distribution ones. We use map layout edit distance from the training layouts as a measure of distribution shift. PLDM demonstrates the best generalization performance. Results are averaged over 3 seeds, shaded area denotes standard error. See Figure 1 for more details on PLDM.

offline data and aim to solve arbitrary tasks specified at test time. HILP [53] proposes learning a distance-preserving representation space such that the distance in that space is proportional to the number of steps between two states, similar to Laplacian representations [69, 70, 73]. Forward-Backward representations [67, 68] tackle this with an approach akin to successor-features [5].

**Optimal Control**, similar to RL, tackles the problem of selecting actions in an environment to optimize a given objective (reward for RL, cost for control). Classical optimal control methods typically assumes that the transition dynamics of the environment are known [7]. This paradigm has been used to control aircraft, rockets, missiles [11] and humanoid robots [35, 57]. When the transition dynamics cannot be defined precisely, they can often be learned [65, 71]. Many RL methods approximate dynamic programming in the context of unknown dynamics [6, 62]. In this work, we use the term RL to refer to methods that either implicitly or explicitly use rewards information to train a policy function, and optimal control for methods that use a dynamics model and explicitly search for actions that optimize the objective.

**The importance of offline data** has been highlighted in works such as ExORL, [76] which demonstrates that exploratory RL data enables off-policy algorithms to perform well in offline RL; however, it only compares exploratory vs. task-specific data, without analyzing which data aspects affect performance. Buckman et al. [12] investigates the data importance for offline RL with rewards. Recently proposed OGBench [51] introduces multiple offline datasets for a variety of goal-conditioned tasks; in contrast, we conduct a more fine-grained analysis of how methods perform in top-down navigation under suboptimal data conditions and generalize to new tasks and layouts. Yang et al. [74] also study generalization of offline GCRL, but focus on reaching out-of-distribution goals. Ghugare et al. [24] study stitching generalization.

## 3    The Landscape of Available Methods

In this section, we formally introduce the setting of learning from state-action sequences without reward annotations and overview available approaches. We also introduce a method we call Planning with a Latent Dynamics Model (PLDM).

### 3.1    Problem Setting

We consider a Markov decision process (MDP) $\mathcal{M} = (\mathcal{S}, \mathcal{A}, \mu, p, r)$, where $\mathcal{S}$ is the state space, $\mathcal{A}$ is the action space, $\mu \in \mathcal{P}(\mathcal{S})$ denotes the initial state distribution, $p \in \mathcal{S} \times \mathcal{A} \rightarrow \mathcal{S}$ denotes the transition dynamics (we only consider the deterministic case), and $r \in \mathcal{S} \rightarrow \mathbb{R}$ denotes the reward function. We work in the offline setting, where we have access to a dataset of state-action sequences $\mathcal{D}$ which consists of transitions $(s_0, a_0, s_1, \ldots, a_{T-1}, s_T)$. We emphasize again that the offline dataset in our setting does not contain any reward information. The goal is, given $\mathcal{D}$, to find a policy $\pi \in \mathcal{S} \times \mathcal{Z} \rightarrow \mathcal{A}$, to maximize cumulative reward $r_z$, where $\mathcal{Z}$ is the space of possible task definitions. Our goal is to make the best use of the offline dataset $\mathcal{D}$ to enable the agent to

Table 1: **Road-map of our generalization stress-testing experiments.** We test 4 offline goal-conditioned methods - HIQL, GCIQL, CRL, GCBC; a zero-shot RL method HILP, and a learned latent dynamics planning method PLDM. ★★★ denotes good performance in the specified experiment, ★★☆ denotes average performance, and ★☆☆ denotes poor performance. We see that HILP and PLDM are the best-performing methods, with PLDM standing out as the only method that reaches competitive performance in all settings.

| **Property** (Experiment section) | HILP | HIQL | GCIQL | CRL | GCBC | PLDM |
|---|---|---|---|---|---|---|
| **Transfer to new environment layouts** (4.8) | ★☆☆ | ★☆☆ | ★☆☆ | ★☆☆ | ★☆☆ | ★★★ |
| **Transfer to a new task** (4.6) | ★★☆ | ★☆☆ | ★☆☆ | ★☆☆ | ★☆☆ | ★★★ |
| **Data efficiency** (4.3) | ★☆☆ | ★★☆ | ★★★ | ★★☆ | ★★☆ | ★★★ |
| **Best-case performance** (4.2) | ★★★ | ★★★ | ★★★ | ★★★ | ★★☆ | ★★★ |
| **Can learn from random policy trajectories** (4.5) | ★★★ | ★☆☆ | ★★★ | ★☆☆ | ★☆☆ | ★★☆ |
| **Can stitch suboptimal trajectories** (4.4) | ★★★ | ★☆☆ | ★★★ | ★☆☆ | ★☆☆ | ★★☆ |

solve a variety of tasks in a given environment with potentially different layouts. During evaluation, unless otherwise specified, the agent is tasked to reach a goal state $s_g$, so the reward is defined as $r_g(s) = \mathbb{I}[s = s_g]$, and $\mathcal{Z}$ is equivalent to $\mathcal{S}$.

## 3.2 Reward-free Offline Reinforcement Learning

In this work, we study methods that solve tasks purely from offline trajectories without reward annotations. Reward-free offline RL methods fall into two categories: goal-conditioned RL and zero-shot methods that treat the task as a latent variable. We evaluate state-of-the-art methods from both categories on goal-reaching, and test zero-shot methods on their ability to transfer to new tasks. The methods we investigate are:

- **GCIQL** [52] – goal-conditioned version of Implicit Q-Learning [33], a strong and widely-used method for offline RL;
- **HIQL** [52] – a hierarchical GCRL method which trains two policies: one to generate subgoals, and another one to reach the subgoals. Notably, both policies use the same value function;
- **HILP** [53] – a method that learns state representations from the offline data such that the distance in the learned representation space is proportional to the number of steps between two states. A direction-conditioned policy is then learned to be able to move along any specified direction in the latent space;
- **CRL** [19] – uses contrastive learning to learn compatibility between states and possible reachable goals. The learned representation, which has been shown to be directly linked to goal-conditioned Q-function, is then used to train a goal-conditioned policy;
- **GCBC** [23, 43] – Goal-Conditioned Behavior Cloning - the simplest baseline for goal-reaching.

## 3.3 Planning with a Latent Dynamics Model

The methods in Section 3.2 are model-free, none explicitly model the environment dynamics. Since we do not assume known dynamics as in classical control, we can instead learn a dynamics model from offline data, similar to [46, 55], which propose a model-based method for goal-reaching using an image reconstruction objective.

We propose a model-based method named Planning with a Latent Dynamics Model (PLDM), which learns latent dynamics using a reconstruction-free SSL objective and the JEPA architecture [39]. At test time, we plan in the learned latent space to reach goals. We opt for an SSL approach that predicts the latents as opposed to reconstructing the input observations [3, 21, 26, 81] motivated by findings that reconstruction yields suboptimal features [2, 42], while reconstruction-free representation learning works well for control [27, 59]. Appendix G provides empirical support: features trained with reconstruction-based methods such as DreamerV3 [26] underperform in test-time planning.

Given agent trajectory sequence $(s_0, a_0, s_1, ..., a_{T-1}, s_T)$, we specify the PLDM world model as:

$$\text{Encoder}: \quad \hat{z}_0 = z_0 = h_\theta(s_0) \tag{1}$$

$$\text{Predictors}: \quad \hat{z}_t^k = f_\theta^k(\hat{z}_{t-1}^k, a_{t-1}), \forall k \in \{1, \ldots, K\} \tag{2}$$

where $\hat{z}_t^k$ is the latent state predicted by predictor $k$ and $z_t$ is the encoder output at step $t$. When $K > 1$, we train an ensemble of predictors for uncertainty regularization at test-time. The training objective involves minimizing the distance between predicted and encoded latents summed over

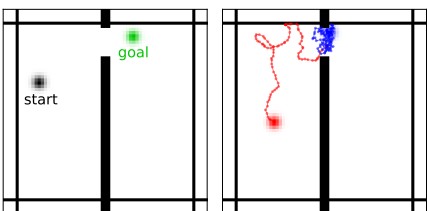

Figure 3: **Left**: The Two-Rooms environment. The agent starts at a random location and is tasked with reaching the goal at another randomly sampled location in the other room using 200 steps or less. Observations are $64 \times 64$ pixels images. **Right**: Examples of trajectories in the offline data. Red: each step's direction is sampled from Von Mises distribution. Blue: each step's direction is sampled uniformly.

Table 2: Performance of tested methods on good-quality data and on data with no trajectories passing through the door. Values are average success rates ($\pm$standard error) across 3 seeds.

| Method | Good-quality data | | | No door-passing trajectories | | |
|---|---|---|---|---|---|---|
| CRL | 89.3 | $\pm$ | 0.7 | 14.7 | $\pm$ | 4.1 |
| GCBC | 86.0 | $\pm$ | 2.0 | 8.4 | $\pm$ | 1.2 |
| GCIQL | 98.0 | $\pm$ | 0.9 | 99.6 | $\pm$ | 0.4 |
| HILP | 100.0 | $\pm$ | 0.0 | 100.0 | $\pm$ | 0.0 |
| HIQL | 96.4 | $\pm$ | 1.3 | 26.3 | $\pm$ | 5.6 |
| PLDM | 97.8 | $\pm$ | 0.7 | 34.4 | $\pm$ | 2.7 |

all timesteps. Given target and predicted latents $Z, \hat{Z}^k \in \mathbb{R}^{H \times N \times D}$, where $H \leq T$ is the model prediction horizon, $N$ is the batch dimension, and $D$ the feature dimension, the similarity objective between predictions and encodings is:

$$\mathcal{L}_{\text{sim}} = \sum_{k=1}^{K} \sum_{t=0}^{H} \frac{1}{N} \sum_{b=0}^{N} \|\hat{Z}_{t,b}^k - Z_{t,b}\|_2^2 \tag{3}$$

To prevent representation collapse, we use a VICReg-inspired [4] objective, and inverse dynamics modeling [40]. We show a diagram of PLDM in Figure 1. See Appendix D.1.1 for details.

**Goal-conditioned planning with PLDM.** In this work, we mainly focus on the task of reaching specified goal states. While methods outlined in Section 3.2 rely on trained policies to reach the goal, PLDM relies on planning. At test time, given the current observation $s_0$, goal observation $s_g$, pretrained encoder $h_\theta$ predictor $f_\theta$, and planning horizon $H$, our planning objective is:

$$\forall k \in \{1, \ldots, K\}: \ \hat{z}_0^k = z_0^k = h_\theta(s_0), \ \hat{z}_t^k = f_\theta^k(\hat{z}_{t-1}^k, a_{t-1}) \tag{4}$$

$$C_{\text{goal}}(\mathbf{a}, s_0, s_g) = \frac{1}{K} \sum_{k=1}^{K} \sum_{t=0}^{H} \|h_\theta(s_g) - f_\theta^k(\hat{z}_t^k, a_t)\| \tag{5}$$

$$C_{\text{uncertainty}}(\mathbf{a}, s_0, s_g) = \sum_{t=0}^{H} \gamma^t \sum_{j=1}^{d} \text{Var}(\{f_{\theta_k}(s_t^k, a_t)_j\}_{k=1}^{K}) \tag{6}$$

$$\mathbf{a}^* = \arg\min_{\mathbf{a}}\{C_{\text{goal}}(\mathbf{a}, s_0, s_g) + \beta C_{\text{uncertainty}}(\mathbf{a}, s_0, s_g)\} \tag{7}$$

$C_{\text{goal}}$ is the goal-reaching objective and $C_{\text{uncertainty}}$ penalizes the model from choosing state-action transitions that deviate from the training distribution, with $\gamma \in [0, 1]$ as the temporal discount. This regularization resembles how GCIQL, HIQL, and HILP use expectile regression to learn policies that remain *in-distribution* with respect to the dataset [34]. See Appendix E for ablations on $C_{\text{uncertainty}}$.

Following the Model Predictive Control framework [45], PLDM re-plans every $i$ interactions with the environment. By default, we use $i = 1$ for all experiments, making PLDM $\sim$ 4x slower than the model-free baselines. The replanning interval $i$ can be increased to accelerate MPC with only a minor loss in performance (see Appendix F). We use MPPI [72] in all our experiments with planning. We note that PLDM is not using rewards, neither explicitly nor implicitly, and should be considered as an optimal control method. We also note that to apply PLDM to a new task, we do not need to retrain the encoder $h_\theta$ and dynamics $f_\theta$, we only need to change the cost in Equation (5). We test this flexibility in Section 4.6, where we invert the sign of the cost to make the agent avoid a given state.

## 4 Every Method Can Excel but Few Generalize

In this section, we conduct thorough experiments testing methods spanning RL and optimal control outlined in Section 3.2 and Section 3.3. We evaluate on navigation tasks where the agent is either a point mass (Section 4.1, Section 4.8) or quadruped (Section 4.7). We generate datasets of varying size and quality and test how a specific data type affects a given method. See Table 1 for overview.

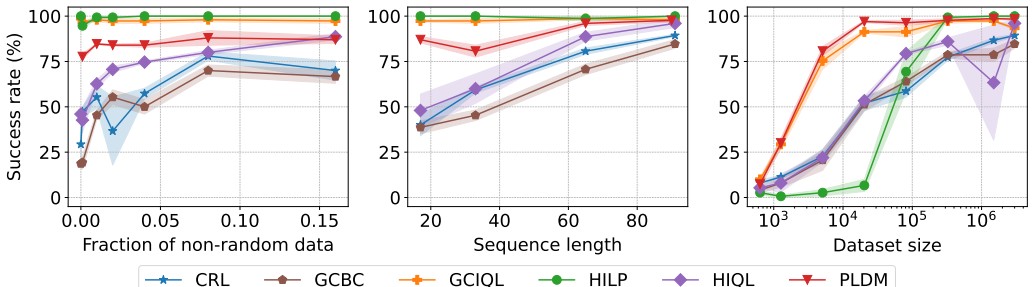

Figure 4: **Testing the selected methods' performance under different dataset constraints.** Values and shaded regions are means and standard error over 3 seeds, respectively. **Left**: To test the importance of the dataset quality, we mix the random policy trajectories with good quality trajectories (see Figure 3). As the amount of good quality data goes to 0, methods begin to fail, with PLDM, GCIQL, and HILP being the most robust ones. **Center**: We measure methods' performance when trained with different sequence lengths. We find that many goal-conditioned methods fail when train trajectories are short, which causes far-away goals to become out-of-distribution for the resulting policy. **Right**: We measure methods' performance with datasets of varying sizes. We see that PLDM and GCIQL are the most sample efficient, and manage to get almost 80% success rate even with a few thousand transitions. See Appendix H for the analysis of statistical significance.

## 4.1 Two-Rooms Environment

We begin with a navigation task called Two-Rooms, featuring a point-mass agent. Each observation $x_t \in \mathbb{R}^{2 \times 64 \times 64}$ is a top-down view: the first channel encodes the agent, the second the walls (Figure 3). Actions $a \in \mathbb{R}^2$ denote the displacement vector of the agent position from one time step to the next, with a norm limit of 2.45. The goal is to reach a randomly sampled state within 200 steps. See Appendix C.2 for more details. This environment allows for controlled data generation – ideal for efficient and thorough experimentation, while still not being too trivial.

**Offline data.** To generate offline data, we place the agent in a random location within the environment, and execute a sequence of actions for $T$ steps, where $T$ denotes the episode length. The actions are generated by first picking a random direction, then using Von Mises distribution with concentration 5 to sample action directions. The step size is uniform from 0 to 2.45. Unless otherwise specified, the episodes' length is $T = 91$, and the total number of transitions in the data is 3 million.

## 4.2 What Methods Excel In-Distribution with a Large High-Quality Dataset?

To get the topline performance of the methods under optimal dataset conditions, we test them in a setting with abundant data, good state coverage, and good quality trajectories long enough to traverse the two rooms. With 3 million transitions, corresponding to around 30,000 trajectories, all methods reach good performance in the goal-reaching task in Two-Rooms (Table 2), with HIQL, GCIQL, HILP, and PLDM nearing 100% success rate.

> **Takeaway**: All methods can perform well when data is plentiful and high-quality.

## 4.3 What Method is the Most Sample-Efficient?

We investigate how different methods perform when the dataset size varies. While our ultimate goal is to have a method that can make use of a large amount of suboptimal offline data, this experiment serves to distinguish which methods can glean the most information from available data. We tried ranges of dataset sizes all the way down to a few thousand transitions. In Figure 4, we see that the model-based PLDM and the model-free GCIQL outperform other model-free methods when data is scarce. In particular, HILP is more data-hungry than other model-free methods but achieves perfect performance with enough data.

> **Takeaway**: PLDM and GCIQL are more sample-efficient than other methods.

## 4.4 What Methods Can Stitch Suboptimal Trajectories?

**Can we learn from short trajectories?** We vary the episode length T during data generation to test whether methods can stitch together short training trajectories to reach long-horizon goals. In real-world scenarios, collecting long episodes is often difficult—especially in open-ended environments—so the ability to learn generalizable policies from short trajectories is crucial. The hardest scenario of Two-Rooms requires the agent to navigate from the bottom left corner to the bottom right corner, and involves $\sim 90$ steps, meaning that with short episode lengths such as 16, the goal is never observed. To succeed, methods must stitch together multiple offline trajectories.

We create datasets with episode length of 91, 64, 32, 16, keeping total transitions number at 3 million. Results in Figure 4 (center) show that with the exception of GCIQL, goal-conditioned model-free methods struggle when trained with shorter episodes. We hypothesize that these methods are limited by dynamic programming on the short transitions, which can be sample-inefficient for stitching many short trajectories to reach far-away goals. In contrast, HILP performs well by learning to follow directions in latent space – even from short episodes. Similarly, PLDM can learn an accurate model from short trajectories and stitch together a plan during test time.

**Can we learn from data with imperfect coverage?** We artificially constrain trajectories to always stay within one room within the episode, and never pass through the door. Without the constraint, around 35% trajectories pass through the door. During evaluation, the agent still needs to go through the door to reach the goal state. This also reflects possible constraints in real-life scenarios, as the ability to stitch offline trajectories together is essential to efficiently learn from offline data. The results are shown in Table 2. We see that HILP and GCIQL achieve perfect performance, while PLDM performance drops but is still higher than the rest of offline GCRL methods. We hypothesize that HILP's latent space structure enables effective stitching, while PLDM retains some performance due to the learned dynamics. With the exception of GCIQL, other model-free GCRL methods fail to learn to compose trajectories across rooms.

> **Takeaway**: When solving the task requires 'stitching', HILP and GCIQL work great. The performance of PLDM drops, but is better than that of most offline model-free GCRL methods.

## 4.5 What Methods Can Learn From Trajectories of a Random Policy?

In this experiment, we evaluate how trajectory quality affects agent performance. In practice, random policy data is easy to collect, while expert demonstrations are often unavailable. Thus, algorithms that can generalize from noisy data are crucial. We create a dataset where actions are sampled uniformly at random, causing agents to oscillate near the starting point. In this setting, the average maximum distance between any two points in a trajectory is $\sim 10$ (when the whole environment is 64 by 64), while when using Von Mises to sample actions as in the case of good quality data, it is $\sim 28$. Example trajectories from both types of action sampling are shown in Figure 3.

We see that HILP, GCIQL and the model-based PLDM outperform the rest of goal-conditioned RL methods in this setting (Figure 4). We hypothesize that because random trajectories on average do not go far, the sampled state and goal pairs during training are close to each other. As a result, faraway goals become out of distribution, and GCRL methods struggle with long-horizon tasks when TD learning fails to bridge distant trajectories. In contrast, PLDM uses the data only to learn the dynamics model, and random policy trajectories are still suitable for the purpose. HILP learns the latent space and how to traverse it in various dimensions – even from random data.

> **Takeaway**: When the dataset quality is low, HILP, GCIQL and PLDM perform better than other offline GCRL methods.

## 4.6 What Methods Can Generalize to a New Task?

To build effective general purpose systems that learn from offline data, we need an algorithm that can generalize to different tasks. So far, we evaluated on goal-reaching tasks. In this experiment, we test generalization to a different task in an environment with the same dynamics: avoiding a chasing agent in the same environment. We compare PLDM and HILP, using models trained on optimal data from Section 4.2 without any further training. In this task, the chaser follows an expert policy along the

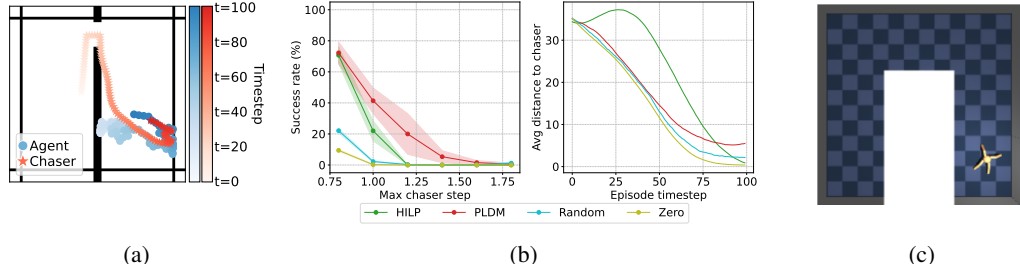

|       |       |       |
| :---: | :---: | :---: |
| (a)   | (b)   | (c)   |

Figure 5: **Zero-shot generalization to the chasing task. (a)** In the chase environment, the blue agent is tasked with avoiding the red chaser. The chaser follows the shortest path to the agent. The observations of the agent remain unchanged: we pass the chaser state as the goal state. The agent has to avoid the specified state instead of reaching it. **(b) Left**: Performance of the tested methods on the chasing task across different chaser speeds, with faster chaser making the task harder. Baselines include agents that take no action ('Zero') and random actions ('Random'). **(b) Right**: Average distance between the agent and chaser agent throughout the episode when chaser speed is 1.0. **(c)** Visualization of ant-umaze environment. The 4-legged ant is tasked with reaching a randomly sampled goal within a u-shaped room.

shortest path to the agent, and we vary its speed to adjust difficulty. The goal of the controlled agent is to avoid being caught. Goal-conditioned methods are excluded from evaluation, as they cannot avoid specific states by design. At each step, the agent observes the chaser's state and selects actions to maintain distance. In PLDM, we invert the sign of the planning objective to maximize distance in the latent space to the goal state. In HILP, we invert the skill direction. We evaluate the success rate - defined as maintaining a distance $\geq 1.4$ pixels over 100 steps. The results are shown in Figure 5b. We also plot average distance between the agents over time in Figure 5b. We see that PLDM performs better than HILP, and is able to evade the chaser more effectively by maintaining greater separation by episode end.

> **Takeaway**: Assuming the environment dynamics remain fixed, PLDM can generalize to tasks other than goal-reaching simply by changing the planning objective.

## 4.7 Extending to a Higher-Dimensional Control Environment.

So far, we have focused on environments with simple control dynamics over a point-mass agent, where actions are 2D displacement or acceleration vectors. We now investigate whether the same trend holds in a setting with more complex control. We choose Ant-U-Maze, a standard state-based environment with a 29-dimensional state space and 8-dimensional action space (Figure 5c). Solving this task with PLDM requires learning non-trivial dynamics that better resemble real-world control.

We collect a dataset using a pretrained directional expert policy from Park et al. [51], generating 5M transitions by resampling a new direction every 10 steps and adding Gaussian noise with standard deviation 1.0 to each action. For evaluation, the quadruped is initialized at the bottom left or right corner, with the goal at the opposite diagonal. Each method is evaluated on 10 trials.

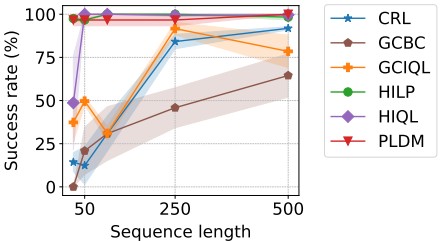

Figure 6: Success rates in Ant-U-Maze for agents trained on trajectories of varying lengths.

As in 4.4, we test the methods' abilities to stitch short training trajectories by using datasets with trajectory lengths 25, 50, 100, 250, and 500 (during evaluation, the start and goal are approximately 200 steps apart). PLDM, HIQL, and HILP outperform other GCRL baselines in trajectory stitching, achieving 100% success rates while other methods fail with shorter trajectories. In contrast to the Two-Rooms experiments, HIQL outperforms GCIQL in this setting. We hypothesize that HIQL's hierarchical structure may particularly benefit high-dimensional ant control: the low-level policy can manage fine-grained joint movements, while the high-level policy can govern overall navigation.

> **Takeaway**: Planning with a latent dynamics model maintains good performance in a standard quadruped maze task, indicating promising generalization to higher control complexity.

## 4.8 What Methods Can Generalize to Unseen Environment Layouts?

In this experiment, we test whether methods can generalize to new obstacle configurations/layouts – a key requirement for general purpose agents, since collecting data for every scenario is infeasible. We introduce a new navigation environment with slightly more complex dynamics and configurable layouts (Figure 2). Building on top of Mujoco PointMaze, [66] layouts are generated by randomly permuting wall locations. Data is collected by randomly sampling actions at each step. Observation includes an RGB top-down view of the maze and agent velocity; actions are 2D accelerations. The goal is to reach a randomly sampled location. See Appendix C.2 for details.

To test generalization to new obstacle configurations, we vary the number of training maze layouts (5, 10, 20, 40) and evaluate on held-out unseen layouts. For models trained on 5 maps, we further analyze how test-time performance degrades as layouts diverge from the training distribution. Figure 8 shows that PLDM generalizes best – even when trained on just 5 maps – while other methods fail. As test layouts become more out-of-distribution, all methods except PLDM degrade in performance. We also evaluate all methods on a single fixed layout and observe 100% success rate across the board (Table 4). Figure 7 and 8 show PLDM's inferred plans and trajectories from different agents. We also investigate HILP's failure to generalize in Appendix I, and show that HILP's learned representation space successfully captures distance between states in mazes seen during training, but fails on unseen mazes.

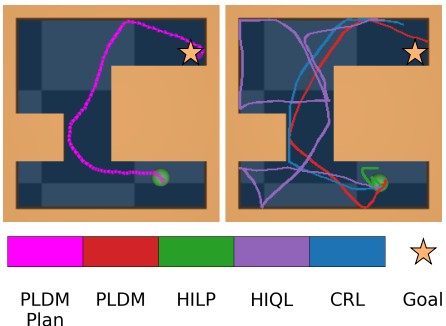

Figure 7: **Left**: Plans generated by PLDM at test time. **Right**: Actual agent trajectories for the tested methods. PLDM is the only method that reliably succeeds on held-out mazes.

> **Takeaway**: The model-based approach enables better generalization to unseen obstacle layouts than model-free methods.

## 5 Conclusion

In this work, we conducted a comprehensive study of existing methods for learning from offline data without rewards, spanning both reinforcement learning and optimal control, aiming to identify the most promising approaches for leveraging suboptimal trajectories. We focus on a set of navigation tasks that present unique challenges due to the need for spatial reasoning, generalization to new layouts, and trajectory stitching. Our findings highlight HILP and PLDM as the strongest candidates, with PLDM demonstrating the best generalization to new obstacle layouts and to a state-avoidance task. We aggregate our experimental results in Table 1. Overall, we draw three main conclusions:

> 1. PLDM exhibits robustness to data quality, a high level of data efficiency, best-of-class generalization to new layouts, and excels at adapting to tasks beyond goal-reaching;
>
> 2. Learning a well-structured latent-space (e.g. using HILP) enables trajectory stitching and robustness to data quality, although it is more data-hungry than other methods;
>
> 3. Model-free GCRL methods are a great choice when the data is plentiful and good quality.

**Future work.** Our findings indicate that learning and planning with latent dynamics models is a promising direction for building general autonomous agents. There are many promising areas for exploration: 1) extending PLDM to more complex domains such as robotic manipulation and partially observable environments; 2) investigating improved dynamics learning methods to mitigate issues like accumulating prediction errors for tasks involving long-horizon reasoning [37]; and 3) improving test-time efficiency, either by backpropagating gradients through the forward model [8] or via amortized planning.

## Acknowledgments

This work was supported through the NYU IT High Performance Computing resources, services, and staff expertise, by the Institute of Information & Communications Technology Planning & Evaluation (IITP) with a grant funded by the Ministry of Science and ICT (MSIT) of the Republic of Korea in connection with the Global AI Frontier Lab International Collaborative Research, by the Samsung Advanced Institute of Technology (under the project Next Generation Deep Learning: From Pattern Recognition to AI) and the National Science Foundation (under NSF Award 1922658), and in part by AFOSR grant FA9550-23-1-0139 "World Models and Autonomous Machine Intelligence" and by ONR MURI N00014-22-1-2773 "Self-Learning Perception through Interaction with the Real World".

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

# Appendix

## Table of Contents

# A   Reproducibility

**Code is available at** https://github.com/vladisai/PLDM

An implementation of our method is provided in the GitHub repository above, which includes environments, data generation, training, and evaluation scripts. Detailed hyperparameters for our method and the baselines are provided in Appendix J.

# B   Visualization of Plans and Trajectories for Diverse PointMaze

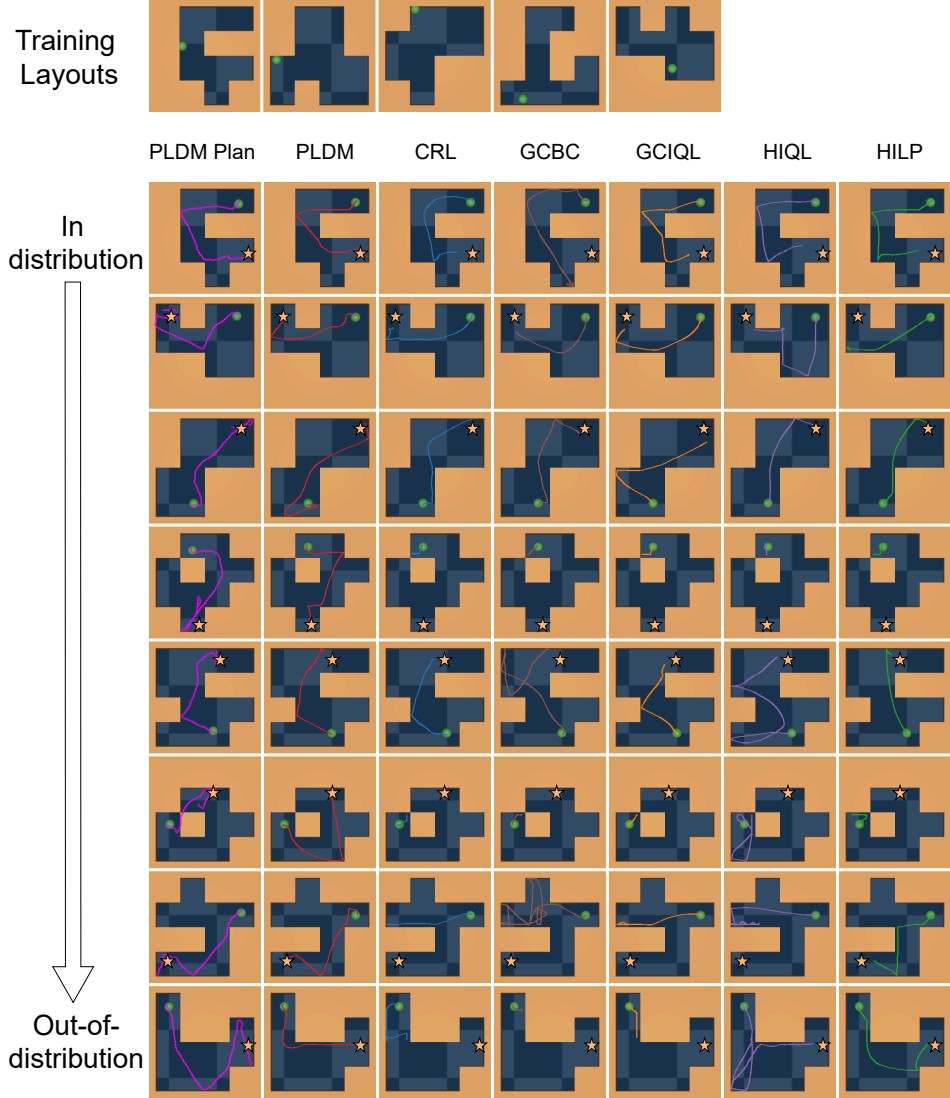

Figure 8: **Top**: The training layouts used in the 5 layout setting. **Middle**: Trajectories of different agents navigating an unseen maze layout towards goal at test time. As the layouts become increasingly out-of-distribution, only PLDN consistently succeeds. Layouts can be represented as a 4x4 array, with each value being either a wall or empty space. The distribution shift is quantified by the minimum edit distance between a given test layout and the closest training layout. The top row corresponds to an in-distribution layout with a minimum edit distance of 0, and with each subsequent row, the minimum edit distance increases by 1 incrementally.

# C Environments and Datasets

## C.1 Two-Rooms Environment

We build our own top-down navigation environment. It is implemented in PyTorch [54], and supports GPU acceleration. The environment does not model momentum, i.e. the agent does not have velocity, and is moved by the specified action vector at each step. When the action takes the agent through a wall, the agent is moved to the intersection point between the action vector and the wall. We generate the specified datasets and save them to disk for our experiments. The datasets generation takes under 30 minutes.

## C.2 Diverse PointMaze

Here, we build upon the Mujoco PointMaze environment [66], which contains a point mass agent with a 4D state vector $(\text{global } x, \text{global } y, v_x, v_y)$, where $v$ is the agent velocity. To allow our models to perceive the different maze layouts, we use as model input a top down view of the maze rendered as $(64, 64, 3)$ RGB image tensor instead of relying on $(\text{global } x, \text{global } y)$ directly.

Mujoco PointMaze allows for customization of the maze layout via a grid structure, where a grid cell can either be a wall or space. We opt for a $4 \times 4$ grid (excluding outer wall). Maze layouts are generated randomly. Only the following constraints are enforced: 1) all the space cells are interconnected, 2) percentage of space cells range from $50\%$ to $75\%$.

We set action repeat to $4$ for our version of the environment.

### C.2.1 Dataset Generation

We produce four training datasets with the following parameters:

| # Transitions | # layouts | # episodes per layout | episode length |
|---|---|---|---|
| 1000000 | 5 | 2000 | 100 |
| 1000000 | 10 | 1000 | 100 |
| 1000000 | 20 | 500 | 100 |
| 1000000 | 40 | 250 | 100 |

Table 3: Details for Diverse PointMaze datasets

Each episode is collected by setting the $(\text{global } x, \text{global } y)$ at a random location in the maze, and agent velocity $(v_x, v_y)$ by randomly sampling a 2D vector with $\|v\| \leq 5$, given that $v_x$ and $v_y$ are clipped within range of $[-5, 5]$ in the environment.

### C.2.2 Evaluation

All the test layouts during evaluation are disjoint from the training layouts. For each layout, trials are created by randomly sampling a start and goal position guaranteed to be at least 3 cells away on the maze. The same set of layouts and trials are used to evaluate all agents for a given experimental setting.

We evaluate agents in two scenarios: 1) How agents perform on test layouts when trained on various number of train layouts; 2) Given a constant number of training layouts, how agents perform on test maps with varying degrees of distribution shift from the training layouts.

For scenario 1), we evaluate the agents on $40$ randomly generated test layouts, $1$ trial per layout.

For scenario 2), we randomly generate test layouts and partition them into groups of 5, where all the layouts in each group have the same degree of distribution shift from train layout as defined by metric $D_{min}$ defined as the following:

Given train layouts $\{L_{\text{train}}^1, L_{\text{train}}^2, ...L_{\text{train}}^N\}$, test layout $L_{\text{test}}$, and let $d(L_1, L_2)$ represents the edit distance between two layouts $L_1$ and $L_2$'s binary grid representation. We quantify the distribution shift of $L_{\text{test}}$ as $D_{\min} = \min_{i \in \{1,2,...,N\}} d(L_{\text{test}}, L_{\text{train}}^{(i)})$.

In this second scenario we evaluate 5 trials per layout, thus a total of $5 \times 5 = 25$ per group.

### C.2.3 Results for Single Maze Setting

Table 4: Results averaged over 3 seeds ± std

| Method | Success rate |
|--------|--------------|
| PLDM | 0.990 ± 0.001 |
| CRL | 0.980 ± 0.001 |
| GCBC | 0.970 ± 0.024 |
| GCIQL | 1.000 ± 0.000 |
| HIQL | 1.000 ± 0.000 |
| HILP | 1.000 ± 0.000 |

### C.3 Ant U-Maze

To investigate whether our findings generalize to environments with more complicated control dynamics, we test the methods on the Ant U-Maze environment [66] with 8-dimensional action space and 29-dimensional state space. Similar to our previous analysis on Two-Rooms, we showcase the trajectory stitching capabilities of different methods.

### C.3.1 Dataset Generation

To collect the dataset, we use a pretrained expert directional policy from Park et al. [51] to generate 5M transitions of exploratory data, where we resample a new direction every 10 steps, and apply Gaussian noise with a standard deviation of 1.0 to every action.

### C.3.2 Evaluation

For evaluation, the quadruped is randomly initialized at either the left bottom or right bottom corner, while the goal location is at the opposite diagonal corner, thus requiring the ant to make 1 turn. Each method is evaluated on 10 trials.

## D Models

For CRL, GCBC, GCIQL, and HIQL we use the implementations from the repository[1] of OGBench [51]. Likewise, for HILP we use the official implementation[2] from its authors.

For the Diverse PointMaze environment, to keep things consistent with our implementation of PLDM (D.1.4), instead of using frame stacking, we append the agent velocity directly to the encoder output.

### D.1 PLDM

#### D.1.1 Objective for collapse prevention

To prevent collapse, we introduce a VICReg-based [4] objective. We modify it to apply variance objective across the time dimension to encourage features to capture information that changes, as

---

[1]https://github.com/seohongpark/ogbench
[2]https://github.com/seohongpark/HILP

opposed to information that stays fixed [61]. The objective to prevent collapse is defined as follows:

$$\mathcal{L}_{\text{var}} = \frac{1}{HD} \sum_{t=0}^{H} \sum_{j=0}^{D} \max(0, \gamma - \sqrt{\text{Var}(Z_{t,:,j}) + \epsilon})$$

$$C(Z_t) = \frac{1}{N-1}(Z_t - \bar{Z}_t)^\top (Z_t - \bar{Z}_t), \ \bar{Z} = \frac{1}{N} \sum_{b=1}^{N} Z_{t,b}$$

$$\mathcal{L}_{\text{cov}} = \frac{1}{H} \sum_{t=0}^{H} \frac{1}{D} \sum_{i \neq j} [C(Z_t)]_{i,j}^2$$

$$\mathcal{L}_{\text{IDM}} = \sum_{t=0}^{H} \frac{1}{N} \sum_{b=0}^{N} \|a_{t,b} - \text{MLP}(Z_{(t,b)}, Z_{(t+1,b)})\|_2^2$$

We also apply a tunable objective to enforce the temporal smoothness of learned representations:

$$\mathcal{L}_{\text{time-sim}} = \sum_{t=0}^{H-1} \frac{1}{N} \sum_{b=0}^{N} \|Z_{t,b} - Z_{t+1,b}\|_2^2$$

The combined objective is a weighted sum of above:

$$\mathcal{L}_{\text{JEPA}} = \mathcal{L}_{\text{sim}} + \alpha \mathcal{L}_{\text{var}} + \beta \mathcal{L}_{\text{cov}} + \delta \mathcal{L}_{\text{time-sim}} + \omega \mathcal{L}_{\text{IDM}}$$

### D.1.2 Ablations of Objective Components

We conduct a careful ablation study over each loss component by setting its coefficient to zero. Two-Room ablations are performed in the optimal setting with sequence length 90, dataset size 3M, and all expert data. Diverse Maze ablations are performed in the 5 training maps setting.

| Ablation | Success rate (Two-Rooms) | Success rate (Diverse Maze) |
|---|---|---|
| – | 98.0 ± 1.5 | 98.7 ± 2.8 |
| var coeff ($\alpha$) | 13.4 ± 9.2 | 11.4 ± 6.5 |
| cov coeff ($\beta$) | 29.2 ± 4.4 | 7.8 ± 4.1 |
| time sim coeff ($\delta$) | 71.0 ± 3.0 | 95.6 ± 3.2 |
| IDM coeff ($\omega$) | 98.0 ± 1.5 | 75.5 ± 8.2 |

### D.1.3 Model Details for Two-Rooms

We use the same Impala Small Encoder used by the other methods from OGBench [51]. For predictor, we use the a 2-layer Gated recurrent unit [14] with 512 hidden dimensions; the predictor input at timestep $t$ is a 2D displacement vector representing agent action at timestep $t$; while the initial hidden state is $h_\theta(s_0)$, or the encoded state at timestep 0. A single layer normalization layer is applied to the encoder and predictor outputs across all timesteps. Parameter counts are the following:

```
total params: 2218672
encoder params: 1426096
predictor params: 793600
```

### D.1.4 Model Details for Diverse PointMaze Environment

For the Diverse PointMaze environment, we use convolutional networks for both the encoder and predictor. To fully capture the agent's state at timestep $t$, we first encode the top down view of the maze to get a spatial representation of the environment $h_\theta : \mathbb{R}^{3 \times 64 \times 64} \to \mathbb{R}^{16 \times 26 \times 26}$, $z^{env} = h_\theta(s^{env})$. We incorporate the agent velocity by first transforming it into planes Expander2D : $\mathbb{R}^2 \to \mathbb{R}^{2 \times 26 \times 26}$, $s^{vp} = \text{Expander2D}(s^v)$, where each slice $s^{vp}[i]$ is filled with $s^v[i]$. Then, we concatenate the expanded velocity tensor with spatial representation along the channel dimension to get our overall representation: $z = \text{concat}(s^{vp}, z^{env}, \dim = 0) \in \mathbb{R}^{18 \times 26 \times 26}$.

For the predictor input, we concatenate the state $s_t \in \mathbb{R}^{18 \times 26 \times 26}$ with the expanded action $\text{Expander2D}(a_t) \in \mathbb{R}^{2 \times 26 \times 26}$ along the channel dimension. The predictor output has the same dimension as the representation: $\hat{z} \in \mathbb{R}^{18 \times 26 \times 26}$. Both the encodings and predictions are flattened for computing the VicReg and IDM objectives.

We set the planning-frequency (Section 3.3) in MPPI to $k = 4$ for this environment.

The full model architecture is summarized using PyTorch-like notations.

```
total params: 53666
encoder params: 33296
predictor params: 20370

PLDM(
    (backbone): MeNet6(
        (layers): Sequential(
            (0): Conv2d(3, 16, kernel_size=(5, 5), stride=(1, 1))
            (1): GroupNorm(4, 16, eps=1e-05, affine=True)
            (2): ReLU()
            (3): Conv2d(16, 32, kernel_size=(5, 5), stride=(2, 2))
            (4): GroupNorm(8, 32, eps=1e-05, affine=True)
            (5): ReLU()
            (6): Conv2d(32, 32, kernel_size=(3, 3), stride=(1, 1))
            (7): GroupNorm(8, 32, eps=1e-05, affine=True)
            (8): ReLU()
            (9): Conv2d(32, 32, kernel_size=(3, 3), stride=(1, 1), padding=(1, 1))
            (10): GroupNorm(8, 32, eps=1e-05, affine=True)
            (11): ReLU()
            (12): Conv2d(32, 16, kernel_size=(1, 1), stride=(1, 1))
        )
        (propio_encoder): Expander2D()
    )
    (predictor): ConvPredictor(
        (layers): Sequential(
            (0): Conv2d(20, 32, kernel_size=(3, 3), stride=(1, 1), padding=(1, 1))
            (1): GroupNorm(4, 32, eps=1e-05, affine=True)
            (2): ReLU()
            (3): Conv2d(32, 32, kernel_size=(3, 3), stride=(1, 1), padding=(1, 1))
            (4): GroupNorm(4, 32, eps=1e-05, affine=True)
            (5): ReLU()
            (6): Conv2d(32, 18, kernel_size=(3, 3), stride=(1, 1), padding=(1, 1))
        )
        (action_encoder): Expander2D()
    )
)
```

### D.1.5 Model Details for Ant-U-Maze

We encode the global $(x, y)$ position using a 2-layer MLP into a 256 dimensional embedding, and concatenate it with the rest of the raw proprioceptive state to make our overall state representation. Our predictor is a 3-layer MLP with ensemble size of 5. During training, variance and covariance regularization is only applied on the part of the representation for $(x, y)$ (first 256 dimensions), since the rest of the proprioceptive state are not encoded and therefore do not collapse.

```
total params: 1080615
encoder params: 9120
predictor params: 1072007

PLDM(
    (backbone): MLPEncoder(
        (globa_xy_encoder): Sequential(
            (0): Linear(in_features=2, out_features=32, bias=True)
            (1): LayerNorm((32,), eps=1e-05, elementwise_affine=True)
            (2): Mish(inplace=True)
            (3): Linear(in_features=32, out_features=256, bias=True)
```

```
            (4): LayerNorm((256,), eps=1e-05, elementwise_affine=True)
        )
        (proprio_encoder): Identity()
    )
    (predictor): MLPPredictor(
        (layers): Sequential(
            (0): Linear(in_features=291, out_features=256, bias=True)
            (1): LayerNorm((256,), eps=1e-05, elementwise_affine=True)
            (2): Mish(inplace=True)
            (3): Linear(in_features=256, out_features=256, bias=True)
            (4): LayerNorm((256,), eps=1e-05, elementwise_affine=True)
            (5): Mish(inplace=True)
            (6): Linear(in_features=256, out_features=283, bias=True)
            (7): LayerNorm((283,), eps=1e-05, elementwise_affine=True)
        )
    )
)
```

# E Effects of Uncertainty Regularization via Ensembles

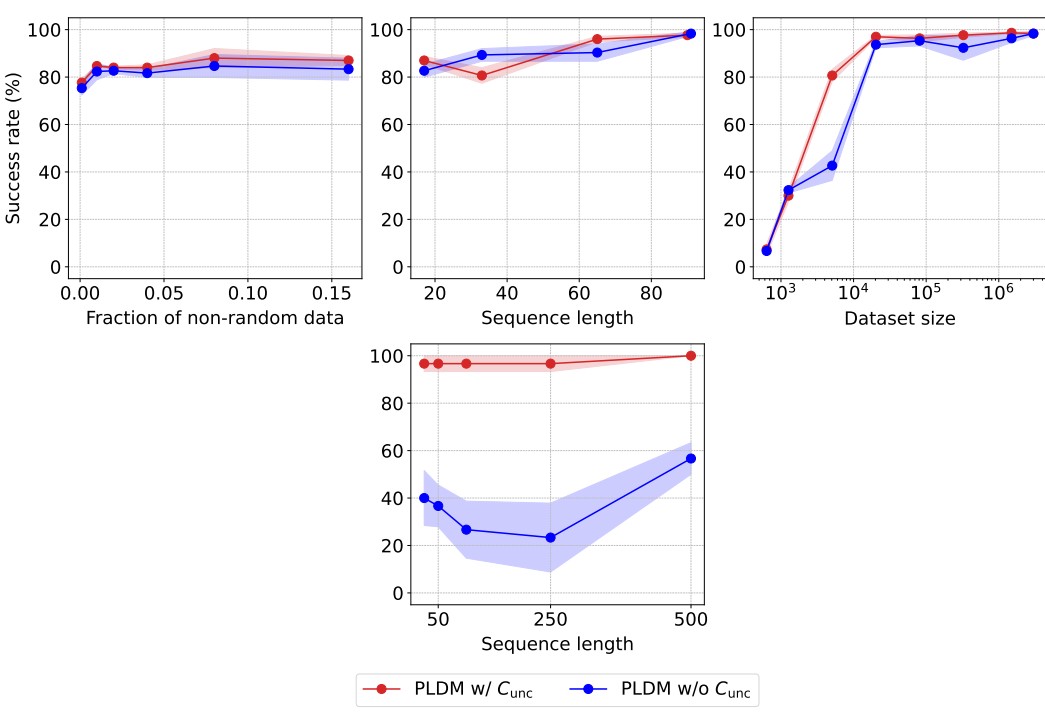

Figure 9: **Top Row**: Two-Rooms environment. **Bottom Row**: AntMaze environment.

# F Analyzing planning time of PLDM

In order to estimate how computationally expensive it is to run planning with a latent dynamics model, we evaluate PLDM, GCIQL, and HIQL on 25 episodes in the Two-Rooms environment. Each episode consists of 200 steps. We record the average time per episode and the standard deviation. We omit HILP, GCBC, or CRL because the resulting policy architecture is the same, making the evaluation time identical to that of GCIQL. HIQL takes more time due to the hierarchy of policies. When replanning every step, PLDM is slower than the policies. However, PLDM can match the latencies of policies by replanning less frequently with negligible performance drop.

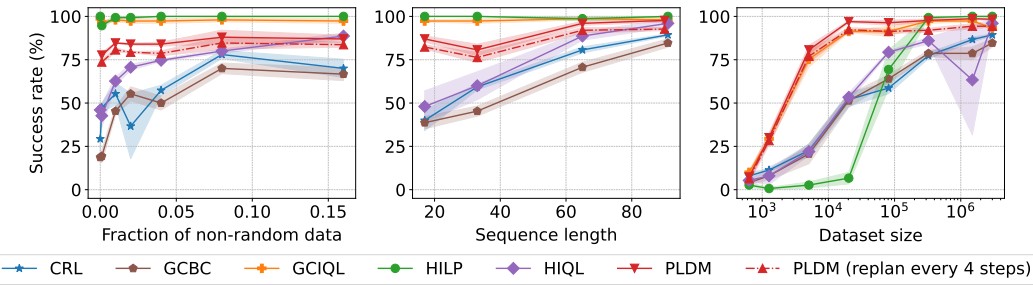

Figure 10: **Comparing PLDM's performance under fixed inference time compute budget on two-rooms.** We see that across two-rooms experiments, PLDM performs only slightly worse when replanning every 4 steps compared to replanning every step.

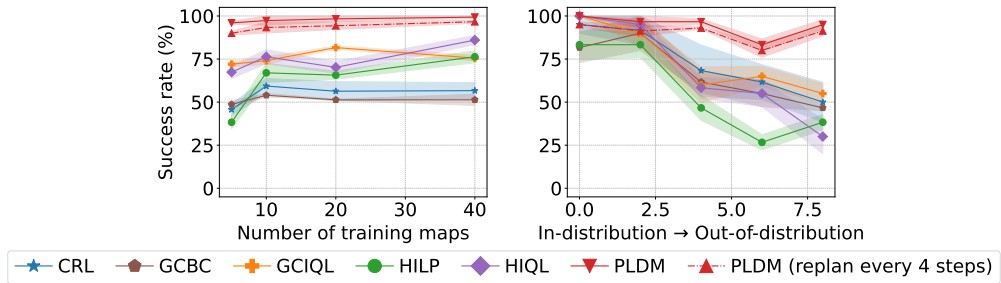

Figure 11: **Evaluating PLDM's performance under fixed inference time compute budget on diverse maze.** Similarly to two-rooms, we only see a minor degradation in performnace

Table 5: Time of evaluation on one episode in Two-Rooms environment. Time is averaged across 25 episodes. PLDM success rates are normalized against the setting that replans every step. PLDM can match the latencies of GCIQL and HIQL by replanning less frequently with neligible cost to performance.

| Method | Replan Every | Time per episode (seconds) | Normalized Success Rate |
|--------|--------------|----------------------------|-------------------------|
| PLDM   | 1            | $16.0 \pm 0.13$            | 1.00                    |
| PLDM   | 4            | $4.8 \pm 0.09$             | 0.95                    |
| PLDM   | 16           | $2.6 \pm 0.07$             | 0.90                    |
| PLDM   | 32           | $2.2 \pm 0.07$             | 0.62                    |
| GCIQL  | –            | $3.6 \pm 0.10$             | –                       |
| HIQL   | –            | $4.0 \pm 0.08$             | –                       |

### F.1 Evaluating PLDM Performance With Adjusted Inference Compute Budget

As we see in Table 5, replanning every 4 steps brings PLDM close to other methods' inference speed. In order to further compare how PLDM compares to other methods when inference time compute is fixed between methods, we run additional evaluations, and report them in Figure 10 and Figure 11.

## G  Extended Baselines: Input Reconstruction Learning and TD-MPC2

In this section, we evaluate reconstruction-based objectives for training the encoder and dynamics model, as well as TD-MPC2 [28]. For reconstruction, we compare two approaches: one based on Dreamer [25], and one that replaces our VICReg objective with pixel reconstruction.

**Dreamer**  We adapt DreamerV3 [26] to our setting by:

- Removing rewards during training and using only the reconstruction loss;
- Omitting policy learning, instead planning via latent representation distance, just like we did for PLDM. Since the representations DreamerV3 uses are discrete, we use KL divergence instead of L2 distance. We note that this is not what DreamerV3 architecture was designed

Table 6: **Comparing reconstruction-based latent-dynamics learning and TDMPC-2 to other baselines.** We test the methods on good-quality data in the two-rooms environment. We see that reconstruction-based methods perform significantly worse than PLDM and other baselines. TD-MPC2 fails to learn altogether due to collapse, and performs poorly even with the added IDM objective.

| Method | Success rate |
|---|---|
| CRL | 89.3 ± 1.2 |
| GCBC | 86.0 ± 4.5 |
| GCIQL | 98.0 ± 0.9 |
| HILP | 100.0 ± 0.0 |
| HIQL | 96.4 ± 3.0 |
| PLDM | 97.4 ± 1.3 |
| DreamerV3 | 24.0 ± 6.9 |
| Reconstruction | 26.2 ± 13.9 |
| TD-MPC2 | 0.0 ± 0.0 |
| TD-MPC2 + IDM | 35.0 ± 0.0 |

for, making this comparison flawed. However, we believe it serves to highlight that compared to DreamerV3, PLDM is designed for a very different setting in terms of available data.

**Reconstruction**   As opposed to Dreamer, this baseline uses the same architecture of the encoder and dynamics as PLDM instead of RSSM, and only replaces the VICReg objective with a reconstruction term. The architecture of the decoder mirrors that of the encoder.

**TD-MPC2**   Like Dreamer models, TD-MPC2 [28] also relies on rewards to learn its representations. In order to adapt TD-MPC2 to our setting, we remove the reward prediction component, and only keep the objective enforcing consistency between the predictor and encoder outputs.

Results are shown in Table 6. We test all methods on the two-rooms environment described in Section 4.1. We use good-quality data, with long trajectories and good transition coverage. We find that pixel observation reconstruction is not a good objective to learn representations, and using it results in poor planning performance. TD-MPC2 collapses altogether, achieving 0% success rate. To prevent collpase, we add inverse dynamics modelling (IDM), which somewhat improves performnace, although it is still far behind other methods. We only tested one seed for TD-MPC2 + IDM.

## H   Analyzing Statistical Significance of Results

In order to analyze whether the results in this paper are statistically significant, we perform Welch's t-test to compare the performance of PLDM to other methods. Because 5 seeds is not enough for statistical tests, we pool results across settings and show results in Table 7. We also run additional seeds for certain selected settings to get a total of 10 seeds per method, and show results of statistical analysis in Table 8. Overall, we see that the results are significant, except for certain settings comparing to HILP and GCIQL, which aligns with our findings.

Table 7: Statistical significance of results (pooled). ✓ means that Welch's t-test showed that PLDM is better than the corresponding method when pooling results across seeds and dataset parameters.

| | Sequence length | Fraction of non-random data | Dataset size | Number of training maps | In-distribution → Out-of-distribution |
|---|---|---|---|---|---|
| CRL | ✓(p=1.68e-03) | ✓(p=8.46e-07) | ✓(p=1.13e-02) | ✓(p=3.43e-10) | ✓(p=4.02e-03) |
| GCBC | ✓(p=4.25e-04) | ✓(p=1.46e-08) | ✓(p=7.26e-03) | ✓(p=3.71e-20) | ✓(p=1.94e-05) |
| GCIQL | ✗ | ✗ | ✗ | ✓(p=1.29e-12) | ✓(p=1.22e-03) |
| HILP | ✗ | ✗ | ✓(p=2.11e-02) | ✓(p=6.54e-06) | ✓(p=4.10e-05) |
| HIQL | ✓(p=2.21e-02) | ✓(p=9.26e-05) | ✓(p=2.39e-02) | ✓(p=2.84e-06) | ✓(p=9.01e-03) |

Table 8: Statistical significance of results for selected datasets. ✓ means that Welch's t-test showed that PLDM is better than the corresponding method results across 10 seeds.

|  | Sequence length 17 | Fraction of non-random data 0% | Dataset size 20312 |
|---|---|---|---|
| CRL | ✓(p=3.82e-11) | ✓(p=1.13e-07) | ✓(p=6.76e-11) |
| GCBC | ✓(p=1.54e-12) | ✓(p=5.75e-10) | ✓(p=1.88e-14) |
| GCIQL | ✗ | ✗ | ✗ |
| HILP | ✗ | ✗ | ✓(p=2.49e-19) |
| HIQL | ✓(p=8.89e-05) | ✓(p=2.83e-07) | ✓(p=2.65e-11) |

# I   Analyzing HILP's Out-of-Distribution Generalization

To understand HILP's poor generalization to out-of-distribution (OOD) maze layouts, we visualize the distance in HILP's learned latent representation space. HILP learns a latent representation $\phi(s)$, such that $\|\phi(s) - \phi(s_g)\|_2$ is equal to the lowest number of transitions needed to traverse from $s$ to $s_g$. We hypothesize that $\phi$ fails to generalize to out-of-distribution maze layouts, resulting in incorrect predicted distances and in the failure of the goal-conditioned policy. We visualize the distances on in-distribution and out-of-distribution layouts for an encoder $\phi$ trained on 5 different layouts in Appendix I. We see that HILP distances are meaningful only on in-distribution layouts, and are very noisy on out-of-distribution layouts. This failure of the latent-space distance to generalize to out-of-distribution layouts confirms our hypothesis, and highlights the strength of PLDM.

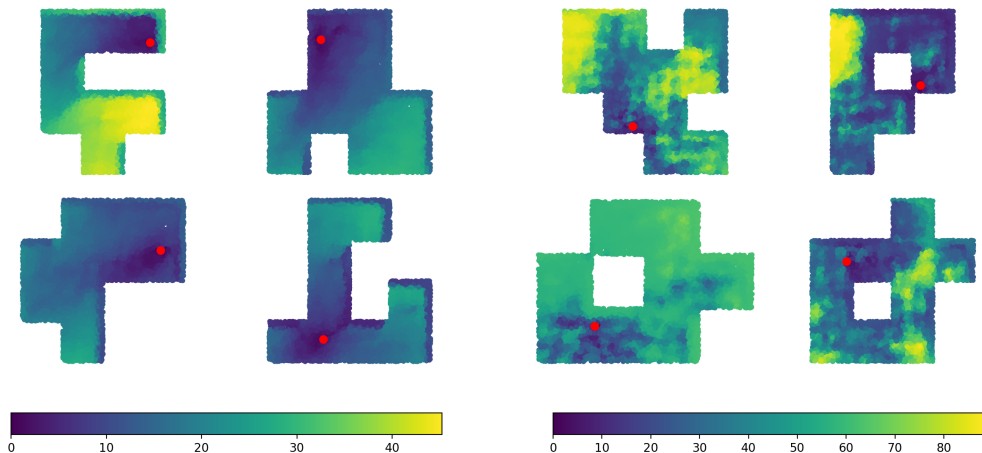

(a) HILP distances on in-distribution maze layouts          (b) HILP distances on out-of-distribution maze layouts

Figure 12: HILP learns representations of states such that distance in representation space between a pair of states is equal to the number of steps needed to go between steps. These plots visualize the distance from the state denoted with a red dot to other states across the maze. (a) We see that on an in-distribution maze, the distances increase smoothly and mostly reflect the number of steps needed to go between states. (b) We see that on an out-of distribution maze, the representation space distances no longer make sense.

# J   Hyperparameters

## J.1   CRL, GCBC, GCIQL, HIQL, HILP

Unless listed below, all hyperparameters remain consistent with default values from OGBench[1] and HILP[2] repositories

### J.1.1 Two-Rooms

To tune hyperparameters, we ran 1 grid search for good quality data, then another grid search for sequence length 17, then tested both on all settings. In grid searches, we searched over learning rate, expectiles (for GCIQL, HILP, and HIQL), and the probability of sampling random goal. For CRL and GCBC, we ended up using default OGBench parameters. The other parameters are described below:

**HILP.** We used learning rate of 3e-4, expectile of 0.7, and skill expectile of 0.7 in all settings.

**GCIQL.** For settings with dataset size of 634 as well as for good quality data, we used expectile 0.7, learning rate 3e-4, BC coefficient of 0.3, and 0 probability of sampling random goal. For all other settings, we used learning rate 3e-4, expectile of 0.7, BC coefficient of 0.003, and probability of sampling a random goal 0.3.

**HIQL.** For sequence length 17 and 33, we used learning rate of 3e-5, expectile of 0.7, AWR temperature of 3.0 (both high and low levels), and probability of sampling a random goal of 0.6. For all other settings, we used learning rate of 3e-4, expectile of 0.7, AWR temperature of 3.0 (both high and low levels), and probability of sampling a random goal of 0.

### J.1.2 Diverse PointMaze

Table 9: Dataset-specific hyperparameters of CRL, GCBC, and HILP for the Diverse PointMaze environment. For HILP, we set the same value for expectile and skill expectile.

| Dataset | CRL | GCBC | HILP | |
| | LR | LR | LR | Expectile |
|---|---|---|---|---|
| 5 layouts | 0.0003 | 0.0003 | 0.0001 | 0.9 |
| 10 layouts | 0.0003 | 0.0001 | 0.0001 | 0.9 |
| 20 layouts | 0.0003 | 0.0001 | 0.0001 | 0.9 |
| 40 layouts | 0.0003 | 0.0001 | 0.0001 | 0.9 |

Table 10: Dataset-specific hyperparameters of GCIQL and HIQL for the Diverse PointMaze environment.

| Dataset | GCIQL | | | | HIQL | | | |
| | LR | Expectile | BC Coeff. | Prob Rand Goal | LR | Expectile | AWR Temp. | Prob Rand Goal |
|---|---|---|---|---|---|---|---|---|
| 5 layouts | 0.0003 | 0.6 | 0.3 | 0.15 | 0.00003 | 0.6 | 1.0 | 0.6 |
| 10 layouts | 0.0003 | 0.6 | 0.3 | 0.15 | 0.0001 | 0.7 | 3.0 | 0.0 |
| 20 layouts | 0.0003 | 0.6 | 0.3 | 0.15 | 0.00003 | 0.6 | 1.0 | 0.6 |
| 40 layouts | 0.0003 | 0.6 | 0.3 | 0.15 | 0.0001 | 0.7 | 3.0 | 0.0 |

### J.1.3 Ant U-Maze

Table 11: Ant-Umaze: dataset-specific hyperparameters for CRL, GCBC, and HILP, parsed from run names.

| Seq Len | CRL | | | GCBC | HILP | |
| | LR | BC Coeff. | Prob Rand Goal | LR | LR | Expectile (skill / actor) |
|---|---|---|---|---|---|---|
| 25 | 3e-5 | 0.1 | 0.3 | 3e-3 | 3e-4 | 0.6 / 0.8 |
| 50 | 3e-5 | 0.1 | 0.3 | 3e-3 | 3e-4 | 0.6 / 0.8 |
| 100 | 3e-4 | 0.1 | 0.0 | 3e-3 | 3e-4 | 0.6 / 0.8 |
| 250 | 3e-4 | 0.1 | 0.0 | 3e-3 | 3e-4 | 0.6 / 0.8 |
| 500 | 3e-4 | 0.1 | 0.0 | 3e-3 | 3e-4 | 0.6 / 0.8 |

Table 12: Ant-Umaze: dataset-specific hyperparameters for GCIQL and HIQL parsed from run names.

| Seq Len | GCIQL | | | | HIQL | | | |
|---|---|---|---|---|---|---|---|---|
| | LR | Expectile | BC Coeff. | Prob Rand Goal | LR | Expectile | AWR Temp. | Prob Rand Goal |
| 25 | 3e-5 | 0.9 | 0.15 | 0.15 | 3e-4 | 0.6 | 3.0 | 0.6 |
| 50 | 3e-5 | 0.9 | 0.15 | 0.15 | 3e-4 | 0.6 | 3.0 | 0.6 |
| 100 | 3e-5 | 0.9 | 0.15 | 0.15 | 3e-4 | 0.6 | 3.0 | 0.6 |
| 250 | 3e-4 | 0.9 | 0.3 | 0.0 | 3e-4 | 0.6 | 3.0 | 0.6 |
| 500 | 3e-4 | 0.9 | 0.3 | 0.0 | 3e-4 | 0.6 | 3.0 | 0.6 |

## J.2 PLDM

### J.2.1 Two-Rooms

The best case setting is sequence length = 91, dataset size = 3M, non-random % = 100, wall crossing % ≈ 35. For our experiments we vary each of the above parameters individually.

Table 13: Dataset-agnostic hyperparameters for Two-Rooms

| Hyperparameter | Value |
|---|---|
| Batch Size | 64 |
| Predictor Horizon ($H$) | 16 |
| Optimizer | Adam |
| Scheduler | Cosine |
| Ensemble size $K$ | 5 |
| $\omega$ | 0 |
| MPPI noise $\sigma$ | 5 |
| MPPI # samples | 500 |
| MPPI $\lambda$ | 0.005 |
| Planner $C_{\text{uncertainty}}$ coeff $\beta$ | 0.0001 |
| Planner $C_{\text{uncertainty}}$ coeff $\gamma$ | 0.9 |

For the dataset specific hyperparameters, we tune the following parameters from Appendix D.1.1:

Table 14: Dataset specific hyperparameters for Two-Rooms

| Dataset | LR | $\alpha$ | $\beta$ | $\delta$ |
|---|---|---|---|---|
| Sequence length = 91 | 0.0007 | 4.0 | 6.9 | 0.75 |
| Sequence length = 65 | 0.0003 | 5.0 | 6.9 | 0.75 |
| Sequence length = 33 | 0.0014 | 3.5 | 6.9 | 0.75 |
| Sequence length = 17 | 0.0028 | 3.0 | 6.9 | 0.75 |
| Dataset size = 634 | 0.0030 | 2.2 | 13.0 | 0.50 |
| Dataset size = 1269 | 0.0010 | 2.2 | 13.0 | 0.50 |
| Dataset size = 5078 | 0.0005 | 2.2 | 13.0 | 0.90 |
| Dataset size = 20312 | 0.0030 | 2.2 | 13.0 | 0.50 |
| Dataset size = 81250 | 0.0010 | 2.2 | 13.0 | 0.50 |
| Dataset size = 325k | 0.0010 | 4.0 | 6.9 | 0.75 |
| Dataset size = 1500k | 0.0010 | 4.0 | 6.9 | 0.75 |
| Non-random % = 0.001 | 0.0007 | 3.9 | 6.9 | 0.74 |
| Non-random % = 0.01 | 0.0007 | 3.9 | 6.5 | 0.19 |
| Non-random % = 0.02 | 0.0007 | 3.9 | 6.5 | 0.72 |
| Non-random % = 0.04 | 0.0007 | 3.9 | 6.5 | 0.65 |
| Non-random % = 0.08 | 0.0007 | 3.9 | 6.5 | 0.24 |
| Non-random % = 0.08 | 0.0007 | 3.9 | 6.5 | 0.24 |
| Wall crossing % = 0 | 0.0007 | 4.0 | 6.9 | 0.75 |

### J.2.2 Diverse PointMaze

Table 15: Dataset-agnostic hyperparameters for Diverse PointMaze

| Hyperparameter | Value |
|---|---|
| Epochs | 5 |
| Batch Size | 128 |
| Predictor Horizon ($H$) | 16 |
| Optimizer | Adam |
| Scheduler | Cosine |
| Ensemble size $K$ | 1 |
| MPPI noise $\sigma$ | 5 |
| MPPI # samples | 500 |
| MPPI $\lambda$ | 0.0025 |

Table 16: Dataset specific hyperparameters for Diverse PointMaze

| Dataset | LR | $\alpha$ | $\beta$ | $\delta$ | $\omega$ |
|---|---|---|---|---|---|
| # map layouts = 5 | 0.04 | 35.0 | 12.0 | 0.1 | 5.4 |
| # map layouts = 5 | 0.04 | 35.0 | 12.0 | 0.1 | 5.4 |
| # map layouts = 20 | 0.05 | 54.5 | 15.5 | 0.1 | 5.2 |
| # map layouts = 40 | 0.05 | 54.5 | 15.5 | 0.1 | 5.2 |

### J.2.3 Ant-U-Maze

Table 17: Dataset-agnostic hyperparameters for Ant-U-Maze

| Hyperparameter | Value |
|---|---|
| Epochs | 5 |
| Batch Size | 64 |
| Predictor Horizon ($H$) | 16 |
| Optimizer | Adam |
| Scheduler | Cosine |
| Ensemble size $K$ | 5 |
| $\alpha$ | 26.2 |
| $\beta$ | 0.5 |
| $\delta$ | 8.1 |
| $\omega$ | 0.58 |
| MPPI noise $\sigma$ | 5 |
| MPPI # samples | 500 |
| MPPI $\lambda$ | 0.0025 |
| Planner $C_{\text{uncertainty}}$ coeff $\beta$ | 1 |
| Planner $C_{\text{uncertainty}}$ coeff $\gamma$ | 0.9 |

Table 18: Dataset-specific hyperparameters for Ant-U-Maze

| Dataset | LR |
|---|---|
| Sequence length = 25 | 0.006 |
| Sequence length = 50 | 0.004 |
| Sequence length = 100 | 0.003 |
| Sequence length = 250 | 0.001 |
| Sequence length = 500 | 0.001 |

### J.3 Further related work

**Offline RL.** This field aims to learn behaviors purely from offline data without online interactions. As opposed to imitation learning [80], offline RL is capable of learning policies that are better than the policy collecting the data. However, a big challenge is preventing the policy from selecting actions that were not seen in the dataset. CQL [36] relies on model conservatism to prevent the learned policy from being overly optimistic about trajectories not observed in the data. IQL [33] introduces an objective that avoids evaluating the Q-function on state-action pairs not seen in the data to prevent value overestimation. MOPO [78] is a model-based approach to learning from offline data, and uses model disagreement to constrain the policy. See [41] for a more in-depth survey.

**Foundation models in RL.** Recently, following the success of NLP, the RL community put a lot of effort into training large sequence models, which sparked dataset collection efforts like Open-X-Embodiment [15] and DROID [31]. Large datasets have enabled training models such as RT-2 [10] and Octo [48]. See [75] for a more extensive survey on the topic.

**Training representations for RL.** Another way to use large amounts of data to improve RL agents is using self-supervised learning (SSL). CURL [38] introduce an SSL objective in addition to the standard RL objectives. Later works also explore using a separate pre-training stage [47, 58, 82]. Zhou et al. [83] show that pre-trained visual representations from DINO [13, 50] can be used to learn a word model for planning.

## K    Discussion of Computational Costs

All experiments require only a single GPU, and take up to 1 day when using an Nvidia V100 GPU. We estimate the total computational cost of the experiments included in the paper and in the research process to be between 500 and 2000 GPU days.

## L    Limitations

All our experiments were conducted in navigation environments, excluding robot manipulation or partially observable settings. However, we argue that the conceptual understanding of the effects of data quality on the investigated methods will hold, as even in the relatively simple setting, we see many recent methods break down in surprising ways. Another limitation lies in the fact that PLDM is about 4 times slower during inference, although in Appendix F we find that even when limited to the same budget of inference compute as model-free methods, PLDM retains most of its performnace.

