# OpenReview forum: "Learning from Reward-Free Offline Data: A Case for Planning with Latent Dynamics Models"
_NeurIPS.cc/2025/Conference — NeurIPS 2025 poster_

### Official Review · Reviewer_saB6 · 2025-06-16

**Clarity:** 3
**Significance:** 2
**Originality:** 3
**Rating:** 5
**Confidence:** 3

**Summary:**

The paper studies how to learn general purpose control policies from offline, reward-free trajectories (focusing specifically on navigation tasks). It compares five recent model-free reinforcement learning approaches (HIQL, GCIQL, CRL, GCBC, HILP) with a new method, Planning with a Latent Dynamics Model (PLDM).  PLDM trains a JEPA-like latent dynamics model from pixel observations using self-supervised objectives, then performs model-predictive control in the latent space. Experiments demonstrate that PLDM is sample efficient, can handle stitching of random trajectories, and generalized best to unseen layouts.

**Questions:**

1. Besides the decoder-based and Dreamer-style ablations, did you test other reconstruction-free SSL objectives (e.g., BYOL, VICReg) within the same planning loop?
2. Can you quantify how much of PLDM’s gain derives from the JEPA encoder versus from having any learned latent model at all? (or provide a rough heuristic as to what you'd expect?)
3. How sensitive is performance to small visual perturbations (Gaussian noise, random crops, mild viewpoint changes)?
4. What are training wall-clock times and peak GPU memory for PLDM versus HILP and HIQL on Two-Rooms and Ant-UMaze?

**Ethical Concerns:**

["NO or VERY MINOR ethics concerns only"]

**Final Justification:**

The author's rebuttal were able to properly address my concerns and make me more confident in accepting this work into NeurIPS.

**Limitations:**

yes

**Paper Formatting Concerns:**

There are very minor typos (these do not affect my score, but should be corrected):

Appendix B.2.3 Table 4: "Method Sucess rate)" should be "Method (Success Rate)"

Appendix C.1.2 In the Table: "Ablation Sucess rate (Two-Rooms) Sucess rate (Diverse Maze)" should be "Ablation Success rate (Two-Rooms) Success rate (Diverse Maze)"

Line 166: "equation Equation (5)" there is a redundant "equation" that can be removed.

Line 645: "coeffient" should be "coefficient"

Line 768 "pefromance" should be "performance"

Line 771: "sigificant" should be "significant"

**Quality:**

3

**Strengths And Weaknesses:**

Strengths:
Systematic analysis of reward-free offline learning is timely and useful for the community; the dataset generator and evaluation protocol will likely be reused. PLDM provides a compelling demonstration that model-based planning can outperform strong model-free methods when data are sparse or misaligned.

Weaknesses:
This paper focuses on navigation tasks. While these tasks are challenging, my main concern is to what extent do these studies replicate to other tasks. Also quite recently there have been other latent planning style architectures for example:

Kong et al. Latent Plan Transformer for Trajectory Abstraction: Planning as Latent Space Inference: https://proceedings.neurips.cc/paper_files/paper/2024/hash/df22a19686a558e74f038e6277a51f68-Abstract-Conference.html

Noh et al. Latent Adaptive Planner for Dynamic Manipulation: https://arxiv.org/abs/2505.03077

These approaches lean more into variational-bayes (MLE-style) approaches for incorporating the latent parameters for planning.
It's unclear if the advantage due to the PLDM method is because of the incorporation of JEPA or if this is an advantage we'd expect to see when incorporating latent parameters into these models in general.

---

> ### Author Rebuttal · Authors · 2025-07-31
>
> ### Weaknesses
>
> > **This paper focuses on navigation tasks. While these tasks are challenging, my main concern is to what extent do these studies replicate to other tasks.**
> ---
>
> While we agree that further experiments are needed to confirm scalability of PLDM to other environments, we would like to reiterate that current experiments with simple environments are design to stress-test specific properties of the existing algorithms for learning from offline reward-free data. While the proposed point-mass environments are simple, they showcase strengths and weaknesses of the existing methods, and uncover unexpected trends. Additionally, we test on ant navigation, an environment that has much more complicated dynamics due to the agent being not a point mass but a 4-legged ant, which showcases PLDM's scalability to complex dynamics.
>
>
> > **Also quite recently there have been other latent planning style architectures for example...**
> ---
>
> Thank you for pointing out these works. We have added them to the related works section.
> However, we couldn't directly compare these to PLDM as [Latent Adaptive Planner for Dynamic Manipulation](https://arxiv.org/abs/2505.03077) does not provide the implementation, and [Latent Plan Transformer for Trajectory Abstraction: Planning as Latent Space Inference](https://proceedings.neurips.cc/paper_files/paper/2024/hash/df22a19686a558e74f038e6277a51f68-Abstract-Conference.html) requires reward signal, although only per-trajectory, making direct comparison impossible.
>
> ### Questions
>
> > **Besides the decoder-based and Dreamer-style ablations, did you test other reconstruction-free SSL objectives (e.g., BYOL, VICReg) within the same planning loop?**
> ---
>
> In addition to the experiments in the paper, we conducted an additional experiment with TDMPC2 architecture and representation learning objective. Because TDMPC2 relies on rewards to learn a Q-function and a policy, we removed those components, but kept the rest unchanged. We found that TDMPC2 without rewards completely collapses, leading to 0% success rate even with good-quality data. We added inverse dynamics learning to TDMPC2, and found that, while avoiding collapse, such model still achieves 0% success rate with good data.
>
> > **Can you quantify how much of PLDM’s gain derives from the JEPA encoder versus from having any learned latent model at all? (or provide a rough heuristic as to what you'd expect?)**
> ---
>
> JEPA architecture only stipulates that the representations and dynamics are learned without reconstruction, and therefore encapsulates TDMPC2 as well. As discussed above, we found that VICReg-based objective performs much better, with TDMPC2 objective failing completely. If we compare JEPA vs non-JEPA approaches, we can look at the comparison between PLDM and reconstruction-based representation learning in Table 6 in the paper. We see that reconstruction-free approaches get less than 30% success rate with good-quality data, while PLDM achieves ~97%.
>
> >  **How sensitive is performance to small visual perturbations (Gaussian noise, random crops, mild viewpoint changes)?**
> ---
>
> We hypothesize that unless the models are trained with such perturbation, they will fail. We are currently running experiments to test if PLDM can work with these perturbations without seeing these transformations during training, we will report the results as soon as they are available.
> ### Formatting issues
> Thank you for pointing out the typos, we have updated the manuscript accordingly.

---

> > ### Comment · Reviewer_saB6 · 2025-08-01
> > **Thank you for the rebuttal**
> >
> > Thank you for your informative responses. After carefully reviewing the paper once again and considering your overall rebuttal I'm leaning towards an accept and will update my score accordingly.

---

### Official Review · Reviewer_2Ba1 · 2025-06-20

**Clarity:** 3
**Significance:** 2
**Originality:** 2
**Rating:** 3
**Confidence:** 4

**Summary:**

This paper investigates the comparative strengths of reinforcement learning (RL) and optimal control methods in the offline setting, specifically when agents must learn from trajectories only. The authors systematically evaluate goal-conditioned RL approaches against a control-based method that trains a latent dynamics model using the Joint Embedding Predictive Architecture (JEPA) for planning. Through experiments on a suite of navigation tasks with varying offline dataset qualities, the study reveals that goal-conditioned model-free algorithms performs well with abundant, high-quality data (especially HILP), while model-based planning exhibits superior generalization to novel environments, enhanced trajectory stitching capabilities, and greater data efficiency. The work provides an empirical analysis of these paradigms under different data and environment conditions, contributing to the understanding of their applicability in reward-free offline learning.

**Questions:**

* Currently, PLDM replans every step, leading to increased computational cost. Could the authors present an experimental analysis demonstrating how performance (e.g., success rate, trajectory quality) varies with different replanning horizons (e.g., replanning every 2nd, 4th, or 8th step)? This investigation would provide crucial insights into the practical deployability and computational trade-offs of the method
* In Sec. 4.5 the authors come to the conclusion that HILP and PLDM generalize better to out of distribution goals. Did the authors investigate the reasons why this is the case?
* Can the authors explain the difference between PLDM's inferred plan and the actual trajectory in Sec. 4.8?

**Ethical Concerns:**

["NO or VERY MINOR ethics concerns only"]

**Final Justification:**

I will update my score since the paper is not a clear reject for me and the authors provided good explanations during the rebuttal. Thus, I update my score to borderline reject. But in my opinion, for such a paper the baselines are all of a similar type with not much variation. Furthermore, the simple setting (i.e. navigation task) cannot really show the strengths and weaknesses of PLDM as good as the authors claim as navigation tasks with such small horizons and in which not much exploration is needed are usually more beneficial for planning-based methods. Therefore, I am not convinced that the experiments "stress-test" PLDM. Thus, I cannot really recommend accepting the paper. But I will not be against a final accept if the AC and the reviewers agree to accept the paper.

**Limitations:**

The authors already describe that their navigation setups limit the overall results of the paper despite arguing directly that this will somehow not be the case. I would add that their choice of baseline algorithms to which PLDM is compared to further limits the generality of the arguments presented.

**Paper Formatting Concerns:**

Section 4.8 refers to Figure 8 and Table 4 in the Appendix which are both crucial for the arguments presented and therefore should be part of the main paper (maybe in a summarized form).

**Quality:**

2

**Strengths And Weaknesses:**

### Strengths
* The paper addresses a large number of questions regarding the performance of different paradigms in goal-conditioned offline learning. The empirical setup explores various factors such as dataset size, data diversity, trajectory stitching, and environment variability, contributing to a comprehensive experimental analysis.
* The paper offers a valuable benchmark of goal-conditioned RL algorithms and a systematic comparison with model-based planning, which is relevant to the community. The insights into the strengths of model-based planning in generalization and data efficiency for these offline settings are impactful and can guide future research.
* The paper is generally well-written and organized, enhancing readability (except Sec. 4.8; see below). The inclusion of section summaries at the end of each major section significantly improves comprehension, making the content accessible to the reader.

### Weaknesses
* Despite answering numerous questions regarding performance, the paper does not adequately delve into why a particular method performs better under specific conditions, limiting the depth of insight. Further investigation into the underlying mechanisms driving observed performance differences would strengthen the analysis.
* The baseline comparisons are somewhat narrow, focusing exclusively on model-free GCRL algorithms (which in my opinion all use an implicit reward with the given goal). I do not share the opinion that goal-conditioned algorithms are the only "reward-free" algorithms and by the title of the paper I would expect a broader range of baseline methods. In Sec. 3.2 the authors mention that they compare to zero-shot methods despite the fact that all baseline methods are goal-conditioned. The absence of other relevant approaches, such as skill-learning methods (e.g., Choreographer) or other unsupervised RL approaches, is a notable limitation. Furthermore, given that the paper's examined method is model-based and emphasizes the advantages of test-time planning, a comparison against a Dreamer-style MBRL algorithm like Director (from Hafner et al., Deep Hierarchical Planning from Pixels) or at least another simple MBRL baseline would have provided a more complete and fair evaluation.
* The tasks utilized in the experiments lack sufficient variation, as acknowledged in the limitations section. While reaching goals may be feasible for MPC in these relatively simple navigation environments, the generalizability of the findings to more complex tasks with larger search spaces (e.g., reaching a diamond in Minecraft) remains questionable. Integrating more diverse and challenging tasks would substantially strengthen the claim of generality.
* The proposed PLDM (Planning with Latent Dynamics Models) replans at every step, which, as shown in Appendix E, results in approximately four times slower computation. The paper would benefit significantly from an investigation into how performance changes with larger replanning horizons (e.g., replanning every 4th step), addressing a practical efficiency concern.
* Section 4.8 exhibits an unusual arrangement. It refers to Figure 2, which is placed at the very beginning of the paper without prior mention or context. Additionally, Section 4.8 relies heavily on Figure 8 and Table 4, both located in the Appendix, to such an extent that they are crucial for understanding the section, not merely supplementary results. This organization impedes flow and clarity.

---

> ### Author Rebuttal · Authors · 2025-07-31
>
> ### Weaknesses
>
> > **1. Despite answering numerous questions regarding performance, the paper does not adequately delve into why a particular method performs better under specific conditions, limiting the depth of insight. Further investigation into the underlying mechanisms driving observed performance differences would strengthen the analysis.**
> ---
>
> Thank you for pointing that out. As you mentioned, the paper is empirical in nature, with limited analysis of particular reasons for failure. We believe that such analysis is beyond the scope of this work. However, we do analyze why HILP fails on layout generalization, and show that the learned distance function used to train the policy does not generalize well to new layouts (section H).
>
> > **2. The baseline comparisons are somewhat narrow, focusing exclusively on model-free GCRL algorithms.**
> ---
>
> We do not solely focus on model-free GCRL algorithms, as HILP is not a goal-conditioned method, and is instead an unsupervised RL method. We also tested Dreamer, and reconstruction-based representation learning, and got poor results even with good data, and therefore didn't test on more settings (see Table 6). Additionally, prompted by questions in this review, we conducted experiments with TDMPC2, and found that due to lack of reward information, TDMPC2 representations completely collapse, leading to 0% success rate.
>
> > **3. The tasks utilized in the experiments lack sufficient variation, as acknowledged in the limitations section.**
> ---
>
> While we agree that further experiments are needed to confirm scalability of PLDM, we would like to reiterate that current experiments with simple environments are design to stress-test specific properties of the existing algorithms for learning from offline reward-free data. While the proposed point-mass environments are simple, they showcase strengths and weaknesses of the existing methods, and uncover unexpected trends. Additionally, we test on ant navigation, an environment that has much more complicated dynamics due to the agent being not a point mass but a 4-legged ant, which showcases PLDM's scalability to complex dynamics.
>
> > **4. The proposed PLDM (Planning with Latent Dynamics Models) replans at every step, which, as shown in Appendix E, results in approximately four times slower computation. The paper would benefit significantly from an investigation into how performance changes with larger replanning horizons (e.g., replanning every 4th step), addressing a practical efficiency concern.**
> ---
>
> We believe there must have been an understanding. We would like to point out that Appendix E tests exactly that, and shows that replanning every 4th step as opposed to every step, retains 95% of the original performance while being only 20% slower than policy-based methods. Please let us know if this is not what you were looking for, we are happy to answer any further questions.
>
> > **5. Section 4.8 exhibits an unusual arrangement.**
> ---
>
> While we agree the figures in the appendix aid in understanding this section, we disagree that they are crucial:
> - Table 4 essentially shows that all the methods get 100% performance, which is exactly what is said in the main text, making the table redundant.
> - Figure 7 is a summarized version of Figure 8 and shows just one layout example. Figure 8 shows trajectories for each method on a separate image, and shows more layouts with different edit distances. While Figure 8 helps in understanding the experiment, we believe Figure 7 is enough to convey the main point. Additionally, Figure 8 is way too large to be able to fit in the main text.
>
> Figure 2 is shown in the beginning of the paper because we believe this is the most interesting result, but we agree that placing it directly in the sction discussing it will be more convenient for the reader. We will move this figure to section 4.8.
>
> ### Questions
>
> > **Currently, PLDM replans every step, leading to increased computational cost. Could the authors present an experimental analysis demonstrating how performance (e.g., success rate, trajectory quality) varies with different replanning horizons (e.g., replanning every 2nd, 4th, or 8th step)? This investigation would provide crucial insights into the practical deployability and computational trade-offs of the method**
> ---
> See response to Weakness 4.
>
> > **In Sec. 4.5 the authors come to the conclusion that HILP and PLDM generalize better to out of distribution goals. Did the authors investigate the reasons why this is the case?**
> ---
>
> We hypothesize that GCRL methods fail on out-of-distribution goals due to trajectory stitching. PLDM can easily stitch trajectories from the training data as the dynamics model predicts one step ahead, and thereofre will learn transitions starting from the same state s but with different actions, e.g. $(s, a_1, s_1')$ and $(s, a_2, s'_2)$ if such transitions are present within any two trajectories in the dataset. HILP works well because it is not technically goal-conditioned, but instead uses latent-space direction to condition the policy (see Figure 1 in [HILP paper](https://arxiv.org/abs/2402.15567)). Covering all existing directions in latent space is easier than all existing goals in the observation space, therefore HILP can reach out of distribution goals as long as they can be reached by following known directions in the latent space. Offline GCRL on the other hand struggles with stitching, as in order to enable stitching, these methods rely on random goal sampling and dynamic programming. The larger the environment, the harder it is to sample reasonable goals to encourage value and policy learning, making such methods fail.
>
> > **Can the authors explain the difference between PLDM's inferred plan and the actual trajectory in Sec. 4.8?**
> ---
>
> The inferred plan is constructed at timestep t=0. We construct a new plan every time we take a step, therefore the resulting executed trajectory may differ from the first planned trajectory due to randomness in the planning algorithm (MPPI samples candidate trajectories randomly) and due to small differences between model predictions and actual environment observations.

---

> > ### Author Response · Authors · 2025-08-03
> >
> > >  **2. The baseline comparisons are somewhat narrow, focusing exclusively on model-free GCRL algorithms.**
> >
> > We'd like to add to the response above and attach results of additional experiments with a popular model-based method TDMPC2. As we mentioned, TDMPC2 relies on rewards, so to adapt it to our reward-free offline setting we had to remove the reward prediction and q-function models and objectives. We found that without those objectives, TDMPC2 representations completely collapse, leading to 0% success rate. Then, to prevent collapse, we added the inverse dynamics modeling objective (predicting $a_t$ given $z_t$ and $z_{t+1}$). We found that the resulting model only gets 35% success rate, which is much lower than PLDM (~97%) and other methods we tested.
> >
> > | Method       | Success rate (%) on two-rooms with good-quality data |
> > | ------------ | ------------------------------------------------ |
> > | PLDM         | 97.8 ± 0.7                                       |
> > | TDMPC2       | 0                                                |
> > | TDMPC2 + IDM | 35                                                 |
> >
> > We believe that comparisons to Dreamer (in Appendix F) and TDMPC2 further highlight the novelty of PLDM, as these methods, although conceptually related to PLDM, are not directly comparable to the methods we study and do not perform well when cast to the reward-free offline setting.
> >
> > We hope that comparisons to unsupervised RL method HILP, as well as comparisons to model-based Dreamer and TDMPC2, address your concern regarding baselines selection. Please let us know if you have any additional questions.

---

> ### Comment · Reviewer_2Ba1 · 2025-08-05
>
> > We do not solely focus on model-free GCRL algorithms, as HILP is not a goal-conditioned method, and is instead an unsupervised RL method.
>
> Based on the information provided in the paper, it is not evident that the used HILP version is not a goal-conditioned method. According to the original publication, HILP can be applied in various settings, including unsupervised, skill-based, zero-shot reinforcement learning, *or* goal-conditioned reinforcement learning. Could the authors clarify which specific variant of HILP was used in their experiments? In particular, does HILP receive the goals (as it seems according to Appendix H)?
>
> >  While the proposed point-mass environments are simple, they showcase strengths and weaknesses of the existing methods, and uncover unexpected trends.
>
> I agree that the selected environments provide a reasonable starting point and illustrate several important capabilities. However, these environments may not adequately reveal the limitations of PLDM. In fact, they might align more closely with the strengths of PLDM—for instance, planning-based methods often face challenges in environments requiring extensive exploration or involving long-horizon tasks. The used environments do not require extensive exploration nor are the involved task long-horizon. Therefore, the generality of the reported results may be more limited than suggested. What is the authors take on this? Does the choice of evaluating on one environment not limit the overall generality of the results?
>
> > We believe there must have been an understanding. We would like to point out that Appendix E tests exactly that, and shows that replanning every 4th step as opposed to every step, retains 95% of the original performance while being only 20% slower than policy-based methods.
>
> Maybe I did not phrase things clearly enough beforehand: In the current experimental setup, PLDM is allocated a computational budget that is approximately four times larger than that of the other baseline methods. From an evaluation fairness perspective, it would be more appropriate to compare the baselines against a version of PLDM that replans every fourth step, thereby matching the computational constraints. As it stands, it is difficult to accurately assess PLDM’s relative performance under equivalent computational budgets, and any conclusions drawn are necessarily speculative.

---

> ### Author Response · Authors · 2025-08-06
>
> ### Is HILP goal-conditioned?
>
> GCRL, CRL, HIQL, GC-IQL methods train a policy that directly accepts current state and the goal state and produces the action, and which therefore can only be applied to the goal-conditioned problems. As is common in the literature, including the paper introducing HILP, we refer to such methods as 'goal-conditioned'. On the other hand, HILP cannot be considered a purely goal-conditioned method, as it is more general and can be used in other settings as we showed in section 4.5, and as was discussed in the original paper. The policy HILP proposes takes as input the current state and a latent variable, which can be derived in different ways, including, but not solely, from a given goal state. The latent variable aims to capture a variety of behaviors, including those not corresponding to reaching a set goal.
>
> We will clarify this distinction in the paper, and say "purely goal-conditioned" when referring to GCRL, CRL, HIQL, and GC-IQL to avoid confusion. Let us know if this addresses your concern.
>
> ### The generality of the reported results is limited
>
> We agree that these results are cannot be used to conclude, once and for all, that PLDM is better than other methods in all cases. The explicitly stated goal of this work is to carefully test methods applicable to reward-free offline data in settings where we have a lot of control over the data. We find that PLDM works well, and therefore attempt to draw attention of the community to this relatively underexplored paradigm of learning latent dynamics and planning. Our findings---for planning in latent dynamics models for diverse environments---fill a gap in the literature and, we believe, would be of great interest to the community. We agree that further extending our investigation to long-horizon tasks and to tasks requiring exploration would be valuable, but our results stand on their own, and we hope to extend this investigation in follow-up work.
>
> That being said, we appreciate your comments regarding the generality of our findings and propose the following changes:
>
> - We will more explicitly state what types of generalization we are testing in each of our experiments. Specifically, we will state that we only test generalization in terms of environment layouts, and in terms of tasks defined on the same state/action space with the same transition function. We will state that other types of generalization, such as generalization to changes in environment dynamics, to noisy observations, etc., are not evaluated for. More specifically:
>     - in line 281, instead of saying "test whether methods can generalize to new environments", we will say "generalize to new obstacle configurations (layouts) within the environment".
>     - In line 288: "to test generalization" will be replaced with "to test generalization to new obstacle layouts"
>     - In line 310: "generalization" will be replaced with "generalization to new environment layouts"
> - In any conclusions we draw from our experiments, we will change the phrasing to specify what kind of generalization we observe. For example:
>     - line 258 "planning with a latent dynamics model can generalize well to a new task.", will be replaced by "Assuming the environment dynamics remain fixed, PLDM can generalize to tasks other than goal-reaching simply by changing the planning objective."
>     - line 305: "The model-based approach enables better generalization to unseen environment variations than model-free methods." will be replaced with "PLDM enables better generalization to unseen obstacle layouts within the environment when compared to model-free methods."
>     - line 314 (the final conclusion box): "Learning a latent dynamics model and using it to plan exhibits robustness to data quality, superior data efficiency, and the best generalization to new layouts and tasks;" will be replaced with "PLDM exhibits robustness to data quality, superior data efficiency, the best generalization to new environment layouts, and excels at adapting to tasks beyond goal-reaching;"
> - In limitations, and at the end of section 3.3, we will reiterate the fact that planning is more computationally expensive (about 4x slower in our experiments) than a single run of the policy trained with RL;
>
> We believe that these changes represent a fair representation of the experimental evidence presented in our work and will make it easier for potential readers to clearly understand relative strengths and weaknesses of RL and optimal control.

---

> > ### Author Response · Authors · 2025-08-06
> >
> > ### Matching the computational constraints
> >
> > We agree that having the computational constraints in mind is important when comparing PLDM to other methods. Following your suggestion, we conducted additional experiments and evaluted PLDM in a variety of settings with replanning every 4 steps. The results will be added to appendix E, and are provided below:
> >
> > Handling noisy data:
> >
> > Method|Expert Data Fraction|Success Rate
> > :---|---:|:---
> > CRL|0.01|55.33 ± 6.36
> > GCBC|0.01|45.33 ± 2.4
> > GCIQL|0.01|64.67 ± 2.91
> > HILP|0.01|99.33 ± 0.67
> > HIQL|0.01|62.67 ± 2.4
> > PLDM|0.01|85.67 ± 1.2
> > PLDM (replan every 4 steps)|0.01|80.67 ± 1.2
> >
> > Stitching short trajectories:
> >
> > Method|Sequence Length|Success Rate
> > :---|---:|:---
> > CRL|17|40.0 ± 6.0
> > GCBC|17|38.67 ± 3.53
> > GCIQL|17|38.0 ± 0.0
> > HILP|17|100.0 ± 0.0
> > HIQL|17|30.67 ± 2.4
> > PLDM|17|87.33 ± 1.76
> > PLDM (replan every 4 steps)|17|82.33 ± 1.2
> >
> > Sample efficiency:
> >
> > Method|Dataset Size|Success Rate
> > :---|---:|:---
> > CRL|5078|22.67 ± 3.71
> > GCBC|5078|20.67 ± 5.7
> > GCIQL|5078|35.33 ± 1.33
> > HILP|5078|2.67 ± 1.76
> > HIQL|5078|22.0 ± 2.31
> > PLDM|5078|80.67 ± 2.6
> > PLDM (replan every 4 steps)|5078|77.0 ± 3.61
> >
> > We see that when replanning every 4 steps, the performance of PLDM drops somewhat, but the overall trend remains.
> >
> > Below the response, we also attach full results across dataset size, sequence length, and expert data fraction experiments.
> >
> > All that being said, we respectfully disagree that---in this case---the comparison is unfair unless the computational resources are equivalent between methods. PLDM, while slower than existing policy-based methods, is not outrageously slow to be completely impractical. With replanning at every step, PLDM needs ~16 seconds for an episode of 200 steps, which amounts to ~80 milliseconds per step, allowing us to run this model at 12.5 hertz if this were a real-time system. This is in line with common robotics applications, with e.g. DROID [1] dataset using 15hz to record action data.
> > Additionally, we would like to highlight that we can easily change the computational budget, and plan longer when compute is available, and plan less when the resources are limited, akin to the recent advances in LLM reasoning [2]. Methods that train a policy and do not do planning lack this flexibility.
> >
> > [1] Khazatsky, Alexander, et al. "Droid: A large-scale in-the-wild robot manipulation dataset."
> >
> > [2] Snell, Charlie, et al. "Scaling llm test-time compute optimally can be more effective than scaling model parameters."
> >
> > ### Full results
> >
> > ###### Expert data fraction
> >
> > Method|Expert Data Fraction|Success Rate
> > :---|---:|:---
> > CRL|0.001|47.33 ± 2.67
> > CRL|0.01|55.33 ± 6.36
> > CRL|0.02|36.67 ± 18.77
> > CRL|0.04|57.33 ± 2.91
> > CRL|0.08|78.0 ± 2.31
> > GCIQL|0.001|49.33 ± 2.91
> > GCIQL|0.01|64.67 ± 2.91
> > GCIQL|0.02|58.0 ± 4.16
> > GCIQL|0.04|68.67 ± 3.71
> > GCIQL|0.08|79.33 ± 1.76
> > HILP|0.001|94.67 ± 1.76
> > HILP|0.01|99.33 ± 0.67
> > HILP|0.02|99.33 ± 0.67
> > HILP|0.04|100.0 ± 0.0
> > HILP|0.08|100.0 ± 0.0
> > HIQL|0.001|42.67 ± 6.77
> > HIQL|0.01|62.67 ± 2.4
> > HIQL|0.02|70.67 ± 1.76
> > HIQL|0.04|74.67 ± 0.67
> > HIQL|0.08|80.0 ± 1.15
> > PLDM|0.001|79.0 ± 1.53
> > PLDM|0.01|85.67 ± 1.2
> > PLDM|0.02|84.67 ± 0.33
> > PLDM|0.04|84.67 ± 0.67
> > PLDM|0.08|88.0 ± 4.04
> > PLDM (replan every 4)|0.001|73.67 ± 1.2
> > PLDM (replan every 4)|0.01|80.67 ± 1.2
> > PLDM (replan every 4)|0.02|79.33 ± 0.33
> > PLDM (replan every 4)|0.04|78.67 ± 1.45
> > PLDM (replan every 4)|0.08|84.67 ± 3.53
> >
> >
> > ###### Sequence length
> >
> > Method|Sequence Length|Success Rate
> > :---|---:|:---
> > CRL|17|40.0 ± 6.0
> > CRL|33|59.33 ± 1.76
> > CRL|65|80.67 ± 1.76
> > CRL|91|89.33 ± 0.67
> > GCIQL|17|38.0 ± 0.0
> > GCIQL|33|58.0 ± 1.15
> > GCIQL|65|80.67 ± 1.33
> > GCIQL|91|93.33 ± 0.67
> > HILP|17|100.0 ± 0.0
> > HILP|33|100.0 ± 0.0
> > HILP|65|98.67 ± 1.33
> > HILP|91|100.0 ± 0.0
> > HIQL|17|30.67 ± 2.4
> > HIQL|33|56.67 ± 3.33
> > HIQL|65|88.67 ± 2.4
> > HIQL|91|96.0 ± 2.31
> > PLDM|17|87.33 ± 1.76
> > PLDM|33|89.33 ± 2.67
> > PLDM|65|96.0 ± 1.15
> > PLDM|90|97.67 ± 0.88
> > PLDM|91|98.33 ± 0.88
> > PLDM (replan every 4)|17|82.33 ± 1.2
> > PLDM (replan every 4)|33|76.33 ± 2.91
> > PLDM (replan every 4)|65|92.0 ± 1.53
> > PLDM (replan every 4)|90|92.67 ± 0.88
> >
> >
> > ###### Dataset size
> > Method|Dataset size|Success Rate
> > :---|---:|:---
> > CRL|1269|11.33 ± 0.67
> > CRL|5078|22.67 ± 3.71
> > CRL|20312|52.0 ± 4.0
> > CRL|81250|58.67 ± 3.33
> > CRL|325000|77.33 ± 1.76
> > GCIQL|1269|12.0 ± 4.16
> > GCIQL|5078|35.33 ± 1.33
> > GCIQL|20312|55.33 ± 1.76
> > GCIQL|81250|74.0 ± 3.06
> > GCIQL|325000|80.67 ± 1.33
> > HILP|1269|0.67 ± 0.67
> > HILP|5078|2.67 ± 1.76
> > HILP|20312|6.67 ± 3.33
> > HILP|81250|69.33 ± 11.22
> > HILP|325000|99.33 ± 0.67
> > HIQL|1269|8.0 ± 5.03
> > HIQL|5078|22.0 ± 2.31
> > HIQL|20312|53.33 ± 2.4
> > HIQL|81250|79.33 ± 0.67
> > HIQL|325000|86.0 ± 2.0
> > PLDM|1269|32.33 ± 1.2
> > PLDM|5078|80.67 ± 2.6
> > PLDM|20312|97.0 ± 0.58
> > PLDM|81250|98.0 ± 0.58
> > PLDM|325000|98.67 ± 0.33
> > PLDM (replan every 4)|1269|28.33 ± 2.6
> > PLDM (replan every 4)|5078|77.0 ± 3.61
> > PLDM (replan every 4)|20312|92.33 ± 1.2
> > PLDM (replan every 4)|81250|91.33 ± 1.2
> > PLDM (replan every 4)|325000|92.0 ± 1.15

---

> > > ### Comment · Reviewer_2Ba1 · 2025-08-06
> > >
> > > Dear authors, thank you for your extensive work and presenting new promising results in such a short time.
> > >
> > > > The policy HILP proposes takes as input the current state and a latent variable, which can be derived in different ways, including, but not solely, from a given goal state.
> > >
> > > Thank you for the explanation of HILP which I appreciate but which was also not needed. I still have a single question: In the experiments of the authors, the *test-time* calculation of the latent vector is based on the goal (i.e. the calculation of the vector is based on Sections 5.2 and 5.3 of the HILP paper)?

---

> ### Author Response · Authors · 2025-08-06
>
> Yes, your understanding is correct, we used the process defined in section 5.2 of the paper introducing HILP.
>
> We did not use test-time planning from section 5.3 in two-rooms as HILP achieves near-perfect performance there even without it. In diverse maze, HILP fails not due to planning complexity, but due to representations failing to capture the distance, as we showed in Appendix H, so we didn't use test-time planning there either.
>
> Let us know if you have any further questions or concerns.

---

### Official Review · Reviewer_WZKp · 2025-07-01

**Clarity:** 3
**Significance:** 3
**Originality:** 2
**Rating:** 4
**Confidence:** 5

**Summary:**

This paper conducts an empirical study to compare model-based planning with goal-conditioned offline RL on learning from offline reward-free datasets. The main method being studied is Planning with Latent Dynamics Models (PLDM), which applies MPPI to a JEPA-style latent dynamics models, using latent distance as cost function to plan for goal-reaching behaviors. The baselines being compared to include GC-IQL, HIQL, HILP and contrastive RL. The main comparisons are done in a proprietary Two-Rooms environment, where a point agent needs to reach a goal. The paper analyzes the behaviors of each method under various levels of data assumptions. Other claims regarding performances in high-dimensional action spaces and generalization to new environments are evaluated on AntMaze and PointMaze respectively. The paper makes numerous findings, a few notable ones listed below:

1. All methods perform well given rich data with good coverage.
2. HILP performs best where significant stitching of trajectories is required, but it is not data efficient.
3. PLDM is data efficient and generalizes best to new tasks and environment designs.

These findings provide valuable insights for researchers and practitioners to advance the field of planning with world models.

**Questions:**

1. In the dreamer baseline in Appendix F, what's the planning cost? Is it pixel distance?
2. Can you include comparisons on more complex domains such as manipulation? An example could be franka kitchen from D4RL, which provides datasets of varying quality and can be cast as a goal-reaching problem.

**Ethical Concerns:**

["NO or VERY MINOR ethics concerns only"]

**Final Justification:**

The authors have addressed my questions regarding the model-based RL baseline, but they have not added experiments in more complex environments. In its current form, the paper offers bits of useful insights into model-based planning. However, it lacks a clear central claim, and the experiments are confined to toy domains. I therefore recommend borderline acceptance, though I would not object to a rejection.

**Limitations:**

The main limitation of this paper, as the authors addressed at the end, is the simplicity of the environments. All the evaluations are in navigation environments with fully observed states. Having results on manipulation problems could be of interest to the broader research community.

**Quality:**

2

**Strengths And Weaknesses:**

**Strengths**
1. The paper conducts extensive experiments using simple, representative methods of each algorithmic class on simple domains.
2. The results are summarized into clear messages and include positive and negative results. This provides a good guideline for researcher and practitioners alike.
3. Specifically, the finding that model-based planning generalizes well to unseen environment designs (e.g. new maze structures), whereas HILP fail to do so, is interesting and opens up potential future research directions

**Weaknesses**
1. The evaluation domains have limited difficulty, and it is unclear how much the claims extend to more complex domains.
2. Some claims presented in the paper are overstated. For example, in Section 4.6, the paper studies generalization to new tasks by inverting the sign of the cost function, converting goal-reaching tasks to goal-avoidance tasks. From this one experiment, the paper claims that PLDM "can generalize well to a new task." This is vastly overstated, as the results might stem from the idiosyncrasies of the environment dynamics.
3. The paper only compares to a single instantiation of model-based planning with a JEPA-style model (a dreamer comparison is included in the appendix in limited domains). Again, it is unclear how much the results transfer to model-based planning with other model architectures.

---

> ### Author Rebuttal · Authors · 2025-07-31
>
> ### Weaknesses
>
> > **1. The evaluation domains have limited difficulty, and it is unclear how much the claims extend to more complex domains.**
> ---
>
> We agree that further experiments are helpful to confirm scalability of PLDM to other environments. However, we would like to reiterate that current experiments with simple environments are design to stress-test specific properties of the existing algorithms for learning from offline reward-free data. While the proposed point-mass environments are simple, they showcase strengths and weaknesses of the existing methods, and uncover unexpected trends. In addition, our results on AntMaze—where the agent is a quadruped rather than a point mass—offer promising early evidence that PLDM can extend to settings with more complex control dynamics.
>
> >  **2. Some claims presented in the paper are overstated. For example, in Section 4.6, the paper studies generalization to new tasks by inverting the sign of the cost function, converting goal-reaching tasks to goal-avoidance tasks. From this one experiment, the paper claims that PLDM "can generalize well to a new task." This is vastly overstated, as the results might stem from the idiosyncrasies of the environment dynamics.**
> ---
>
> Thank you for pointing this out. We will tone down this claim. However, we'd like to emphasize that this experiment serves to showcase that PLDM can be adapted to an entirely different task just by changing the planning objective -- an ability that GCRL methods lack, no matter how simple the task is.
>
>
> >  **3. The paper only compares to a single instantiation of model-based planning with a JEPA-style model (a dreamer comparison is included in the appendix in limited domains). Again, it is unclear how much the results transfer to model-based planning with other model architectures.**
> ---
>
> While it's true that we only consider a single instantiation of a model-based algorithm, we did try Dreamer-style reconstruction, and found performance very poor even with good-quality data. Additionally, prompted by this review, we conducted experiments with TDMPC2, and found that it performs poorly or collapses completely.
> Existing model-based methods like Dreamer and TDMPC2 heavily rely on the provided reward information in the data, and fail when the data is not available. Thus, comparing PLDM to these methods is not an apples to apples comparison. We are considering the reward-free offline setting, which, while very important, hasn't been explored in the literature as much as learning from rewards was.
>
> ### Questions
>
> > **In the dreamer baseline in Appendix F, what's the planning cost? Is it pixel distance?**
> ---
>
> We used the latent representation space distance, just like we did for PLDM.
>
> >  **Can you include comparisons on more complex domains such as manipulation? An example could be franka kitchen from D4RL, which provides datasets of varying quality and can be cast as a goal-reaching problem.**
> ---
>
> We are running experiments with manipulation, but we are afraid that this will take longer than the duration of the rebuttal. Also, as mentioned above, we believe the existing experiments already support the message we are conveying in this paper.

---

> > ### Comment · Reviewer_WZKp · 2025-08-01
> >
> > How do you compute the latent distance-to-goal in the Dreamer baseline? Dreamer is a recurrent architecture. Do you encoder the goal as a sequence of length 1?

---

> > > ### Author Response · Authors · 2025-08-01
> > >
> > > You are right, to get the representation of the target, we use DreamerV3 RNN to encode the sequence of 1 state (see equation 1 in DreamerV3 manuscript [1]). This gives us $z_\mathrm{target}$. This amounts to running the encoder $q_\phi(z_t | h_t, x_t)$ with zero-initialized $h_t$ (initial empty RNN state), and $x_t$ set to the target observation.
> > >
> > > Then, given current state $s_0$ and the proposed sequence of actions $a_0, \ldots, a_t$, we unroll the sequence model and the dynamics recurrently, and get the RNN state $h_t$ and the dynamics prior $z_t$.
> > >
> > > Finally, we get a measure of distance between the target and predicted prior state, which we then minimize using planning. To measure that distance, we calculate $\mathrm{KL}(z_\mathrm{target} \Vert z_t)$.
> > >
> > > Let us know if this answers your question.
> > > Thank you for pointing this out, we will clarify this in the manuscript.
> > >
> > > [1] Hafner, Danijar, et al. "Mastering diverse domains through world models." arXiv preprint arXiv:2301.04104 (2023).

---

> > > > ### Comment · Reviewer_WZKp · 2025-08-01
> > > >
> > > > I'm not fully convinced this is the right way to evaluate Dreamer. Notably, the KL divergence is not a distance -- it's not symmetric. Moreover, the latents contain information about the history, so it's not valid to compare the latent for a history of observations and the one-step latent of the goal image. That said, I'm not sure if there is an apples-to-apples comparison to Dreamer. The closest is to train a goal-conditioned Dreamer with a binary reward for reaching the goal. But this is no longer model-based planning. I would encourage the authors to clarify the limitation of this baseline in the paper. The rest of the paper is well-supported by empirical evidence. Therefore, I will maintain my assessment of the paper.

---

> > > > > ### Author Response · Authors · 2025-08-03
> > > > >
> > > > > We agree that there's no direct 'apples-to-apples' comparison to Dreamer, as it was not designed for model-based planning. The protocol we described is the best we could do with the Dreamer architecture without re-training it with a goal-conditioned reward. The aim of the experiment was to show that the widely used MBRL method Dreamer cannot be directly applied to the offline reward-free setting, highlighting the novelty of PLDM. To make the experiment clearer for the readers of this paper, *we will add the detailed protocol description and the conceptual limitation of the comparison to the section discussing this baseline.*
> > > > >
> > > > > We'd like to add that, prompted by the reviews, we conducted experiments with TDMPC2, and now have additional results. Similar to Dreamer, TDMPC2 relies on reward signal. Therefore, we had to remove the reward and q-function components of the model and objective, but kept the rest unchanged. We found that without the reward signal, TDMPC2 completely collapses, achieving 0% success rate. We then added inverse dynamics modeling to prevent collapse, and got somewhat better performance of 35% success rate on two-rooms with good data. This is much worse than PLDM with ~97%.
> > > > >
> > > > > | Method       | Success rate on two-rooms with good-quality data |
> > > > > | ------------ | ------------------------------------------------ |
> > > > > | PLDM         | 97.8 ± 0.7                                       |
> > > > > | TDMPC2       | 0 (collapsed representations)                                              |
> > > > > | TDMPC2 + IDM | 35                                                 |
> > > > >
> > > > > We believe that *comparisons to Dreamer and TDMPC2 further highlight the novelty of PLDM*, as these methods, although conceptually related to PLDM, are not directly comparable to the methods we study and do not perform well when cast to the reward-free offline setting.
> > > > >
> > > > > Thank you for taking the time to provide the feedback and for helping us make this paper better!

---

### Official Review · Reviewer_XygG · 2025-07-01

**Clarity:** 3
**Significance:** 2
**Originality:** 2
**Rating:** 3
**Confidence:** 4

**Summary:**

In this paper, different RL and control-based methods are evaluated on a suite of navigation tasks with offline dataset of varying quality and a latent dynamics model using JEPA is proposed for planning. Extensive experiments are conducted to show the advantages and disadvantages of different approaches, especially the ability of the latent dynamics model in handling suboptimal offline data and adapting to diverse environment configurations.

**Questions:**

1. While in the literature, many benchmarks with varying data quality, e.g., D4RL, have been proposed, it is unclear why these two new tasks were proposed as an alternative testbed, especially considering these two navigation tasks are actually simple. Can D4RL also be used in the experiments as the testbed?
2. As can be seen from the results, except for generalization, the performance of PLDM is not competitive, and in many cases, HILP even performs better. Then what are the advantages of PLDM compared with HILP? Moreover, as mentioned, in the literature, the use of latent dynamics model for planning is actually not new, and more discussions and comparisons among these latent-state based approaches should be reported.
3. As is well-known, the performances of model-based approaches rely heavily on the accuracy of the learned model, then how can we guarantee the accuracy of the model of PLDM?  Is it also the reason that these approaches are only compared in the two simple navigation tasks as more complex the task is, the less accuracy of the learned model may be.
4. Experiments on some large-scale environments are welcome to show the scalability of PLDM.

**Ethical Concerns:**

["NO or VERY MINOR ethics concerns only"]

**Final Justification:**

All the authors' claims on not using some more complex environments to verify PLDM are that the adopted simple tasks can be used to better generate data with various qualities. It is obviously true that the simple tasks can facilitate the stress-test of specific properties of the existing algorithms, but a better performance in simple tasks does not mean it can still achieve good results on more complex domains. Without the results on tasks such as the Kitchen tasks and the manipulation environments, where HILP performs good, it is hard to say that planning with a latent dynamics model proves to be a strong approach for handling suboptimal offline data and adapting to diverse environment configurations. So, my evaluation and the score of the paper are unchanged.

**Limitations:**

Yes

**Paper Formatting Concerns:**

I do not notice any major formatting issues in this paper.

**Quality:**

2

**Strengths And Weaknesses:**

Strengths:
Extensive experiments are conducted to show the effectiveness of the proposed latent dynamics model along with a full discussion of the advantages and disadvantages of different RL and optimal control-based methods. Furthermore, two new navigation environments with granular control over the data generation process are proposed as the testbed.

Weakness:
1) Many works have been proposed to show relationships between RL and optimal-control based approaches, and latent dynamics models are also commonly used for planning in the literature, the contribution of the paper is incremental;
2) In many cases, the performances, except for the generalization ability of the proposed latent dynamics, are not competitive compared with existing methods;
3) These approaches are only verified on two simple navigation tasks, the scalability of the proposed model in some real-world applications is unclear.

---

> ### Author Rebuttal · Authors · 2025-07-31
>
> ### Weaknesses:
>
> > **1. Many works have been proposed to show relationships between RL and optimal-control based approaches, and latent dynamics models are also commonly used for planning in the literature, the contribution of the paper is incremental;**
> ---
>
> We agree that the relationship between RL and optimal control has been explored since the very inception of RL, and we do not claim novelty in either being the first to investigate the relationship between RL and control, or to propose planning. However, we argue that our proposed approach is novel because it considers the planning with latent-dynamics in the reward-free setting. Previous works that use planning, such as TD-MPC2, use reward signal. PLDM on the other hand doesn't use rewards and therefore is more flexible in terms of what data it can use, and what tasks it can be applied to downstream. Furthermore, our work includes comprehensive empirical evaluations across multiple datasets with varying data quality, which, as reviewers saB6, 2Ba1, and WZKp noted, provide valuable insights into the strengths and limitations of existing methods in the reward-free offline regime.
> We would be happy to clarify or expand on any part of our contribution, and if there are specific related works we’ve overlooked, we would gladly cite them upon suggestion.
>
> > **2.  In many cases, the performances, except for the generalization ability of the proposed latent dynamics, are not competitive compared with existing methods;**
> ---
>
> Aside from generalization, PLDM exhibits stronger robustness to data quality that GCRL methods across all experiments. Additionally, although HILP has competitive performance with PLDM on sequence length and data quality, PLDM is significantly more data efficient.
>
> > **3. These approaches are only verified on two simple navigation tasks, the scalability of the proposed model in some real-world applications is unclear.**
> ---
>
> While we agree that extending our expeirments to more complicated environments would be valuable, we believe we already provide strong enough evidence with experiments spanning both visual and proprioceptive domains, as also mentioned by reviewer 3zum. These simple environments are designed in a way to have a minimally complex setting to stress-test methods for offline reward-free data, and we believe these environments achieve this goal -- we see existing methods fail in unexpected ways. Additionally, granular control of data generation afforded by the simple environments we consider would be much harder in more complicated environments like manipulation.
>
> ### Questions:
>
> > **1. Why another benchmark, and not D4RL?**
> ---
>
> While D4RL is indeed a great benchmark for offline RL, it is not suitable for the reward-free context, as each environment there only has one task. For example, the navigation tasks considered in D4RL only have one goal state in the maze. Our Two-Rooms and Diverse Maze environments, while simple, sample the goal location randomly, making it much more challenging.
>
> > **2. As can be seen from the results, except for generalization, the performance of PLDM is not competitive, and in many cases, HILP even performs better. Then what are the advantages of PLDM compared with HILP? Moreover, as mentioned, in the literature, the use of latent dynamics model for planning is actually not new, and more discussions and comparisons among these latent-state based approaches should be reported.**
> ---
>
> We appreciate the reviewer’s question. While HILP demonstrates strong performance in stitching and on low-quality data, PLDM performs competitively in these settings and brings distinct advantages that make it a valuable contribution. In particular, PLDM is significantly more data-efficient (Section 4.3), and it shows stronger generalization to both unseen tasks (Section 4.6) and unseen layouts (Section 4.8), which is crucial for scalable offline learning.
> We agree that latent dynamics models for planning have been explored in prior work. However, to the best of our knowledge, PLDM is the first end-to-end method to systematically apply this paradigm in the offline, reward-free setting. We appreciate the reviewer’s suggestion to expand the discussion of related latent-state planning approaches and will include a more detailed comparison in the final version of the paper.
>
> >  **3. As is well-known, the performances of model-based approaches rely heavily on the accuracy of the learned model, then how can we guarantee the accuracy of the model of PLDM? Is it also the reason that these approaches are only compared in the two simple navigation tasks as more complex the task is, the less accuracy of the learned model may be.**
> ---
>
> PLDM relies on model ensembles to measure dynamics model uncertainty (see Eq. 6 in the paper). While this is not a perfect measure of model accuracy, it can serve as its proxy during planning, and limit considered actions to ones where the dynamics model is accurate. We already show this can work on ant navigation (see section 4.7), an environment with relative complex dynamics. Overall, we believe that uncertainty estimation will enable PLDM to scale to even more complicated dynamics.
>
> > **4. Experiments on some large-scale environments are welcome to show the scalability of PLDM.**
> ---
> We agree that further experiments are helpful to confirm scalability of PLDM to other environments. However, we would like to reiterate that current experiments with simple environments are design to stress-test specific properties of the existing algorithms for learning from offline reward-free data. While the proposed point-mass environments are simple, they showcase strengths and weaknesses of the existing methods, and uncover unexpected trends. In addition, our results on AntMaze—where the agent is a quadruped rather than a point mass—offer promising early evidence that PLDM can extend to settings with more complex control dynamics.

---

> > ### Author Response · Authors · 2025-07-31
> >
> > >  **2. As can be seen from the results, except for generalization, the performance of PLDM is not competitive, and in many cases, HILP even performs better. Then what are the advantages of PLDM compared with HILP? Moreover, as mentioned, in the literature, the use of latent dynamics model for planning is actually not new, and more discussions and comparisons among these latent-state based approaches should be reported.**
> > ---
> > To address your concern regarding limited novelty compared to existing methods that use planning with the latent space, we conducted further experiments with TDMPC2, an existing method that uses planning with a latent dynamics model. Unlike PLDM, TDMPC2 heavily relies on reward signal during training, so we had to remove reward and value prediction components of TDMPC2 to adapt it to our setting. In our experiments, we found that TDMPC2 latent space completely collapses without reward signal, resulting in 0% success rate even with good-quality data on two-rooms environment. To prevent collapse, we added inverse dynamics modeling (IDM). We then grid-searched hyperparameters, such as learning rate, momentum coefficient, and IDM loss coefficient, and found that the best-performing TDMPC2 with IDM gets 35% success rate on good data, which falls behind PLDM with 97.8% success rate.
> >
> > We believe that this additional experiment, combined with our experiments with Dreamer in appendix F, serve as **evidence that for learning from reward-free offline data, PLDM is much better than the existing model-based methods, making it a valuable contribution to the field**.
> >
> > Let us know if our responses addresses your concerns. We remain available to answer any further questions.

---

### Official Review · Reviewer_3zum · 2025-07-03

**Clarity:** 3
**Significance:** 3
**Originality:** 2
**Rating:** 5
**Confidence:** 4

**Summary:**

The authors focus on the problem of learning control from a set of reward-free trajectories (i.e. offline data). In particular, the authors focus on a comparison of offline reward-free RL algorithms with the optimal control (OC) method, utilizing a learned JEPA model that the authors refer to as PLDM. The authors compare the algorithms along the following axes: transfer to new environments, transfer to new tasks, data efficiency, best-case performance, learning from a random policy, and trajectory stitching. The experiments are performed on a custom "two-room" task, AntMaze and PointMaze tasks. The authors show results in which that the proposed model either outperforms or performs similarly to the best baseline in all tested categories.

**Questions:**

- Any hypothesis how would the model-based method perform with a smaller action repeat?

- How central is JEPA to the performance here? What would happen with the TD-MPC2 model instead of JEPA?

**Ethical Concerns:**

["NO or VERY MINOR ethics concerns only"]

**Final Justification:**

This paper offers a joint analysis of OC and RL in the context of reward-free RL. Although the technical novelty is limited, I find the number of experiments satisfactory, and most of my reservations concern the description of experiments and the use of strong language in describing the results - the authors have promised to adjust the camera-ready version accordingly. As such, I recommend acceptance.

**Limitations:**

Yes, though I believe that the authors should tone down their interpretation of the results

**Quality:**

2

**Strengths And Weaknesses:**

Overall I believe this is an interesting and valuable manuscript. My biggest issue is with the description of the "generalization" experiments and with a strong wording of the claims.

Strengths:

- Reward-free learning is an important direction
- The paper is clearly written
- Experiments on both vision and proprioceptive environments

Weaknesses:

- I think that the authors should be more explicit about what "new task" and "new environment" actually mean in terms of the dynamics of the MDP. For example, changing robot embodiment (e.g. ant -> humanoid or even ant -> ant with longer legs) is different than displacing walls in a maze. I think that this is especially important considering that the method proposed explicitly learns a dynamics model. To the best of my understanding, these perturbations have local impact on the dynamics model.

- In general, I think that a lot of the presented experiments favor methods that explicitly learn a dynamics model (which is policy agnostic) rather than a Q-function (which is conditioned on a policy). Performing experiments where there are limited/local perturbations to the dynamics is fine, but in my opinion the authors should tone down their interpretation of the results (e.g. "(...) model-based planning demonstrates superior generalization to novel layouts, better trajectory stitching, and greater data efficiency")

- To the best of my understanding, the hyperparameters for the baselines are not tuned (the authors use the default values from OGBench), while the proposed method uses separate hyperparameter settings per environment. If the baselines were tuned via the same procedure as the proposed method the results would be more convincing

- Why not use any model-based RL baselines? For example, TD-MPC2 [1] was shown to work in an offline setup and also uses a learned model, but there are a variety of other papers

Nitpicks:

- I would highly prefer to have metrics instead of colored stars in Table 1 - it is hard to say what "average performance" or other actually mean

- There is a lot going on in Figure 1

- I do not like that the authors motivate their work with a broad comparison of RL and OC in the offline setting, but end up using a single model learning OC algorithm which is also a method proposed in this work. I think that the introduction would work better if authors just said that they propose an OC/model learning method which they will benchmark against RL

- The first contribution relating to the proposed environments and data generating process could be documented better. Where is the "granular control" described?

---

> ### Author Rebuttal · Authors · 2025-07-31
>
> ### Weaknesses
>
> > **I think that the authors should be more explicit about what "new task" and "new environment" actually mean in terms of the dynamics of the MDP.**
> ---
>
> Thank you for pointing this out. We will add this description to the paper.
> By new task, we mean any task defined on the same environment, with the same state/action space and dynamics. We picked avoidance task as it's a natural task for navigation, and showcases flexibility of PLDM which GCRL methods don't have.
> In terms of new environment, we only consider layout perturbations, and assume that there exists a distribution of layouts, some samples from which are seen during training. During inference, unseen samples are used for evaluation.
>
> > **In general, I think that a lot of the presented experiments favor methods that explicitly learn a dynamics model (which is policy agnostic) rather than a Q-function (which is conditioned on a policy).**
> ---
>
> In this work, we aim to investigate what methods can learn well from reward-free offline data. To do well with this data, methods need to be able to generalize to new tasks. We disagree that this favors dynamics models specifically, but we do agree that this setting points out an important limitation in most commonly used value-based methods in RL: the policy collecting the data may be completely irrelevant to the downstream task, which impedes Q-learning. In this paper, we argue that, instead of Q-learning, a promising way to tap into reward-free offline data is learning a dynamics model and planning (PLDM), as it's better at handling low-quality data, and can be used to solve different tasks after training.
>
> > **To the best of my understanding, the hyperparameters for the baselines are not tuned (the authors use the default values from OGBench), while the proposed method uses separate hyperparameter settings per environment. If the baselines were tuned via the same procedure as the proposed method the results would be more convincing **
> ---
>
> We do tune baseline hyperparameters for diverse maze, see appendix I.1.2 for the exact valus.
> For two rooms, we tuned hyperparameters for the good-quality data, and used them throughout the rest of the experimetns. We are currently running hyperparameter tuning for experiments with sequence length set to 17, as tuning all methods hyperparameters for each individual setting will be prohibitively expensive. The experiments are still running for GCBC, HIQL, CRL. We didn't tune HILP as it already achieves perfect performance. We did observe a boost in performance for GCIQL. We will update results of GCIQL and other methods as the experimetns complete.
>
> | Method   | Success Rate Before Tuning | Success Rate After Tuning (single seed) |
> | -------- | -------------------------- | --------------------------------------- |
> | GCIQL    | 34.8  +-  4.4              | 48.0                                    |
> | PLDM<br> | 83.5  +- 3.7               | x                                       |
>
> > **Why not use any model-based RL baselines? For example, TD-MPC2 [1] was shown to work in an offline setup and also uses a learned model, but there are a variety of other papers**
> ---
>
> We ran additional experiments to test TDMPC2 performance in our setting. Since we are in the reward-free offline setting, we removed the reward and q-function learning components from TDMPC2, but kept the rest of the architecture unchanged. We found that TDMPC2 collapses without additional reward supervision and achieves 0% success rate.
>
>
> ### Nitpicks
>
> > **I would highly prefer to have metrics instead of colored stars in Table 1 - it is hard to say what "average performance" or other actually mean**
> ---
>
> Thank you for pointing this out. We will redo Table 1 to have stars denote percentile performance of the methods.
>
> > **There is a lot going on in Figure 1**
> ---
>
> Thank you for pointing this out, we will simplify it by removing the results part of the figure.
>
> > **I do not like that the authors motivate their work with a broad comparison of RL and OC in the offline setting, but end up using a single model learning OC algorithm which is also a method proposed in this work. I think that the introduction would work better if authors just said that they propose an OC/model learning method which they will benchmark against RL**
> ---
>
> Thank you for this thoughtful comment. We agree that clarifying the framing in the introduction would improve the paper and will revise it to more clearly state that we are introducing a novel OC/model-learning method. That said, the broader motivation of the work remains to explore the relative strengths and weaknesses of RL and OC in the offline reward-free setting. PLDM was proposed specifically because existing OC methods do not work under these conditions. To support this comparison, we have included Dreamer as an additional OC baseline and added results for TD-MPC2 in this rebuttal.
>
> > **The first contribution relating to the proposed environments and data generating process could be documented better. Where is the "granular control" described?**
> ---
>
> Thank you for pointing this out. By granular control, we mean the flexibility the proposed environment affords in adjusting data collection properties, such as agent behavior (purely random or biased for one direction), state coverage (limit trajectories to one room), episode lengths, etc. The TwoRooms environment is also GPU-accelerated, meaning that we can quickly collect new datasets and experiment with many variations of the datasets. We will add this discussion to the paper in the sections describing the environments.
> ### Questions
>
> > **Any hypothesis how would the model-based method perform with a smaller action repeat?**
> ---
>
> Smaller action repeat, e.g. 2 instead of 4, will mean that covering the same distance will require twice the number of steps. This will require a longer planning horizon, which will in turn mean that the model error will accumulate more, leading to worse performance. On the other hand, we hypothesize that long-horizon tasks can be solved more easily with planning if action repeat is increased.
>
> > **How central is JEPA to the performance here? What would happen with the TD-MPC2 model instead of JEPA?**
> ---
>
> While PLDM and TD-MPC2 use different methods for learning the representation, TD-MPC2 can be considered to be a JEPA, as it learns representations and the dynamics by predicting in the latent space and does not use reconstruction.
> As other reviewers also pointed out the relevance of TDMPC2, we ran additional experiments to test its performance. Since we are in the reward-free offline setting, we removed the reward and q-function learning components from TDMPC2, but kept the rest of the architecture unchanged. We found that TDMPC2 collapses without additional reward supervision and achieves 0% success rate.

---

> ### Comment · Reviewer_3zum · 2025-07-31
> **Thank you for the rebuttal**
>
> Overall, I think this is an interesting work, and I am generally leaning towards acceptance. I am satisfied with the number of experiments and their results. However, the descriptions of what is being done were sometimes lacking - I understand that authors are ready to improve these in the final version.
>
> Given above, the only thing holding me back is the use of strong language throughout the paper that sometimes feels like overclaiming the benefits of using classic control versus RL. I would be much more confident in the paper if the authors slightly toned down the commentary of the results. Can authors propose something (e.g. toning down the 'generalization' claims, more open discussion of computational costs)?

---

> ### Author Response · Authors · 2025-08-01
>
> Thank you for your feedback! We are glad you find the experiments and their results convincing.
>
> We appreciate your point regarding the strength of our claims. We reviewed the claims in the paper, and propose the following changes in view of your feedback:
> - We will explicitly state what types of generalization we are testing in each of our experiments. We will state that we only test generalization in terms of environment layouts, and in terms of tasks defined on the same state/action space with the same transition function. We will state that other types of generalization, such as generalization to changes in environment dynamcs, to noisy observations, etc, are not evaluated for. More specifically:
> 	- in line 281, instead of saying "test whether methods can generalize to new environments", we will say "generalize to new obstacle configurations (layouts) within the environment".
> 	- In line 288: "to test generalization" will be replaced with "to test generalization to new obstacle layouts"
> 	- In line 310: "generalization"  will be replaced with "generalization to new environment layouts"
> - In any conclusions we draw from our experiments, we will change the phrasing to specify what kind of generalization we observe. For example:
> 	- line 258 "planning with a latent dynamics model can generalize well to a new task.", will be replaced by "Assuming the environment dynamics remain fixed, PLDM can generalize to tasks other than goal-reaching simply by changing the planning objective."
> 	- line 305: "The model-based approach enables better generalization to unseen environment variations than model-free methods."  will be replaced with "PLDM enables better generalization to unseen obstacle layouts within the environment when compared to model-free methods."
> 	- line 314 (the final conclusion box): "Learning a latent dynamics model and using it to plan exhibits robustness to data quality, superior data efficiency, and the best generalization to new layouts and tasks;"  will be replaced with  "PLDM exhibits robustness to data quality, superior data efficiency, the best generalization to new environment layouts, and excels at adapting to tasks beyond goal-reaching;"
> - In limitations, and in the end of section 3.3, we will reiterate the fact that planning is more computationally expensive (about 4x slower in our experiments) than a single run of the policy trained with RL;
>
> We believe that these changes will make the claims more in line with the experimental evidence presented in our work, and will make it easier for potential readers to clearly understand relative strengths and weaknesses of RL and OC.
>
> Let us know if these changes address your concerns.

---

> > ### Comment · Reviewer_3zum · 2025-08-01
> >
> > Implementing these changes will address my concerns - I will adjust my score to reflect this.

---

> > > ### Author Response · Authors · 2025-08-03
> > >
> > > We are glad to hear that the proposed changes address your concern. Thank you for providing feedback and helping us make this paper better!
> > >
> > > As promised, we also attach results for baselines hyperparameter tuning for the two-rooms environment, with sequence length 17. As we said, we didn't tune HILP as it already achieves perfect performance. We did observe a boost in performance for GCIQL and HIQL, but no improvement for CRL and GCBC. Even with the improvement of GCIQL and HIQL, PLDM remains the best-performing method in this setting by a large margin. Results:
> > >
> > > | Method | Success Rate Before Tuning | Success Rate After Tuning (single seed) |     |
> > > | ------ | -------------------------- | --------------------------------------- | --- |
> > > | GCIQL  | 34.8  +-  4.4              | 48.0                                    |     |
> > > | HIQL   | 33.2  +- 4.6               | 52.0                                    |     |
> > > | CRL    | 37.6  +- 9.1               | 34.0                                    |     |
> > > | GCBC   | 34.0  +- 7.8               | 34.0                                    |     |
> > > | PLDM   | 83.5  +- 3.7               | x                                       |     |
> > > | HILP   | 100.0  +- 0                | x                                       |     |

---

### Note · Authors · 2025-08-12

We would like to thank all the reviewers for helping us make improve this paper.
We were delighted that the reviewers appreciated our work:
- The experiments provide valuable, comprehensive analysis useful to the community (reviewers saB6, 2Ba1, WZKp);
- The dataset and the benchmark are likely to be reused (saB6);
- The paper is clearly written (reviewer 3zum), with results summarized into clear messages (reviewer WZKp).

## Key findings and additional environments (reviewers XygG, WZKp, 2Ba1, saB6)

We appreciate reviewers' suggestions for experiments in additional environments. We agree that this may be helpful in making our stress test even more unambiguous, but our findings already paint a clear picture. We chose environments carefully, with the intent of isolating specific effects, and our results successfully highlight the differences in methods' performances, and while we agree that more experiments are often better, the results in **our manuscript stand on their own and address the question we set out to answer**. With world models gaining in popularity, the community still lacks a clear understanding of their advantages and disadvantages in different settings, and our findings take an important first step in answering this question. As such, **we are confident that the results in our submission are timely and would be of great interest to the NeurIPS community, and we hope you agree and consider supporting that our work be presented at NeurIPS 2025.**

Thank you!

## Remarks on experiments requested by reviewers and provided in the rebuttal


### Selection of baselines and comparison to model-based methods (reviewers 3zum, XygG, WZKp, 2Ba1, saB6)

We conducted experiments with TDMPC2, and show the results below. We found that **TDMPC2 collapses** without reward-prediction. To prevent collapse, we added inverse dynamics loss, but still observed poor performance. **We will add these results to appendix F.**

Method|Success rate (two-rooms, good-quality data)
---|---
PLDM|97.8±0.7
TDMPC2|0 (collapsed)
TDMPC2+IDM|35

### Computational cost of PLDM (reviewer 2Ba1)

We conducted additional experiments with replanning every 4 steps to keep computational budget fixed across methods, and found that the overall trends remain unchanged, even though the performance drops by a few percent. Further, even with replanning at every step, PLDM can work at 12.5hz, making it applicable in most robotics settings. **We will add the new results to appendix E**.

---

### Decision · Program_Chairs · 2025-09-17

**Decision:**

Accept (poster)

**Comment:**

This paper presents a systematic comparison between model-free reinforcement learning (RL) and model-based planning for solving tasks from reward-free offline data. The authors propose Planning with Latent Dynamics Models (PLDM), a method that learns a latent dynamics model with a Joint Embedding Predictive Architecture (JEPA) and uses it for planning. The study evaluates various approaches on a suite of navigation tasks, analyzing performance across different factors like data quality, data efficiency, and generalization to new environments and tasks.

Strengths:
- The reviewers praised the paper for tackling an important and timely problem in offline RL. There was a strong consensus that the experimental evaluation is extensive and valuable, providing a comprehensive analysis that is useful to the community.
- The clarity of the writing and the organization of the paper were also highlighted as significant strengths, with reviewers noting that the results are summarized into clear messages .

Weaknesses:
- The primary concern, shared by multiple reviewers, is the limited complexity and variety of the evaluation domains. The experiments are largely confined to simple navigation tasks, which raises questions about the scalability of the findings and the proposed method (PLDM) to more complex, real-world problems

Recommendation: This paper has a mixed reception/ While the concerns about the limited evaluation environments are valid, the paper provides a well-executed and valuable empirical study on a very relevant problem. The authors' careful, controlled experiments successfully "stress test" different methods and yield clear insights. Their commitment to refining their claims in the final version significantly strengthens the paper. Balancing the valuable empirical contributions against the limited scope, the paper leans towards acceptance.